# The Overcooked Generalisation Challenge:
# Evaluating Cooperation with Novel Partners in Unknown Environments Using Unsupervised Environment Design

**Constantin Ruhdorfer**                                    *constantin.ruhdorfer@vis.uni-stuttgart.de*
*Collaborative Artificial Intelligence*
*University of Stuttgart, Germany*

**Matteo Bortoletto**                                       *matteo.bortoletto@vis.uni-stuttgart.de*
*Collaborative Artificial Intelligence*
*University of Stuttgart, Germany*

**Anna Penzkofer**                                          *anna.penzkofer@vis.uni-stuttgart.de*
*Collaborative Artificial Intelligence*
*University of Stuttgart, Germany*

**Andreas Bulling**                                         *andreas.bulling@vis.uni-stuttgart.de*
*Collaborative Artificial Intelligence*
*University of Stuttgart, Germany*

**Reviewed on OpenReview:** *https://openreview.net/forum?id=K2KtcMlW6j*

## Abstract

We introduce the Overcooked Generalisation Challenge (OGC) – a new benchmark for evaluating reinforcement learning (RL) agents on their ability to cooperate with unknown partners in unfamiliar environments. Existing work typically evaluated cooperative RL only in their training environment or with their training partners, thus seriously limiting our ability to understand agents' generalisation capacity – an essential requirement for future collaboration with humans. The OGC extends Overcooked-AI to support dual curriculum design (DCD). It is fully GPU-accelerated, open-source, and integrated into the `minimax` DCD benchmark suite. Compared to prior DCD benchmarks, where designers manipulate only minimal elements of the environment, OGC introduces a significantly richer design space: full kitchen layouts with multiple objects that require the designer to account for interaction dynamics between agents. We evaluate state-of-the-art DCD algorithms alongside scalable neural architectures and find that current methods fail to produce agents that generalise effectively to novel layouts and unfamiliar partners. Our results indicate that both agents and curriculum designers struggle with the joint challenge of partner and environment generalisation. These findings establish OGC as a demanding testbed for cooperative generalisation and highlight key directions for future research. We open-source our code[1].

## 1 Introduction

Developing computational agents capable of collaborating with each other has emerged as a key challenge in artificial intelligence (AI) research (Dafoe et al., 2020). Recent years have seen considerable advances in developing cooperative reinforcement learning (RL) agents (Stone et al., 2010; Hu et al., 2020; Choudhury et al., 2020; Ding et al., 2024) and several benchmarks were proposed to evaluate their generalisation abilities (Samvelyan et al., 2019; Bard et al., 2020). However, these benchmarks typically treat generalisation

---

[1]Our project web-page is accessible at `https://collaborative-ai.org/publications/ruhdorfer25_tmlr/`.

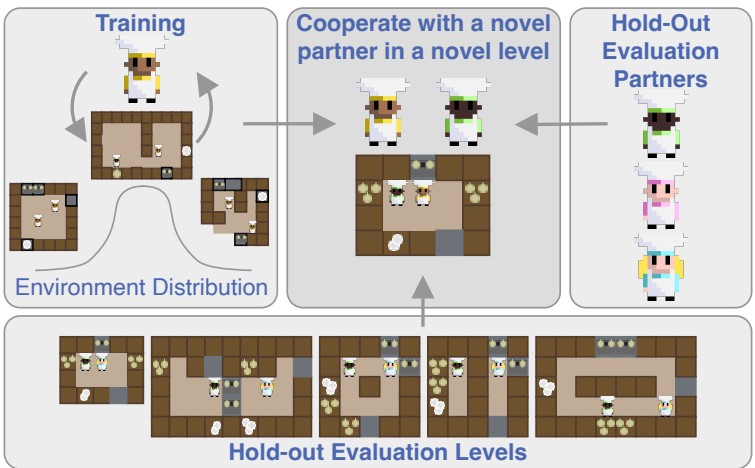

Figure 1: In the Overcooked Generalisation Challenge (OGC), during training, agents can access a generator that outputs new training environments. During evaluation, agents are presented with a novel environment and an unknown partner to cooperate with.

to novel environments (Cobbe et al., 2019) and novel partners (Hu et al., 2020; Carroll et al., 2019) as distinct challenges. Yet, future human-AI collaboration will require agents to *generalise along both axes simultaneously.* For instance, an autonomous robot assisting in a disaster response team must coordinate with ever-changing human partners in unfamiliar, dynamic environments.

Overcooked-AI (Carroll et al., 2019) has emerged as one of the most popular benchmarks for evaluating zero-shot coordination. Nonetheless, agents are typically trained and evaluated on a few fixed layouts (Strouse et al., 2021; Yang et al., 2022; Zhao et al., 2023; Yu et al., 2023; Wang et al., 2024). This common practice limits the benchmark's ability to assess generalisation in two key ways: First, agents may overfit to specific spatial configurations, interaction bottlenecks, or object placements seen during training, without acquiring coordination strategies transferable to novel environments. Second, agents may implicitly adapt to their training partners' behaviour patterns on known layouts, but this does not test their ability to infer intent or adapt dynamically to the behaviour of unknown partners. Crucially, these two challenges are closely linked: coordination strategies often depend on the structure of the environment. For example, in layouts with narrow passages or asymmetrical tasks, agents need carefully coordinate their movements or take on complementary roles. Testing how well agents generalise to new partners in just one environment overlooks this connection. Similarly, evaluating agents only in new environments with familiar partners misses the challenge of adapting to different behaviours. Real-world teamwork requires agents to generalise jointly over both axes.

To address these limitations, we introduce the *Overcooked Generalisation Challenge* (OGC) – a novel cooperation benchmark based on Overcooked-AI to evaluate RL agents' ability to collaborate with unknown partners and in novel environments (see Figure 1). In contrast to prior work that augments a fixed set of test environments for training Jha et al. (2025), we introduce an unsupervised environment design (UED) approach (Dennis et al., 2020), situated within the broader Dual Curriculum Design (DCD) framework, to procedurally generate a large set of diverse training layouts. DCD refers to a framework in which a generator (designer) and a level selector (curator) co-evolve to construct a curriculum tailored to the agent's learning progress (Jiang et al., 2021a). This enables us to evaluate generalisation in a more challenging and realistic way, where agents encounter entirely novel environments without prior exposure. As such, the OGC is the first benchmark to combine UED with multi-agent zero-shot cooperation, thus bridging two previously separate lines of research.

We evaluate trained agents on a suite of human-authored test layouts across three coordination settings: self-play, cross-play, and ad-hoc teamwork. We show that OGC presents a significant challenge for current UED algorithms and scalable neural architectures. Among the evaluated methods, only PAIRED (Dennis et al., 2020) combined with a Soft Mixture-of-Experts (SoftMoE) policy (Obando-Ceron et al., 2024) achieves partial generalisation. Our results reveal a key limitation: current dual curriculum design (DCD) methods

with hand-crafted or weak procedural designers fail to generate sufficiently diverse and structured training levels, and thus struggle in complex design spaces like Overcooked. By contrast, methods with learned environment generators, such as PAIRED, show better adaptability. These findings highlight the need for joint-curriculum methods that combine effective partner generalisation with adaptive curriculum generation. Our contributions are as follows:

1. We introduce the Overcooked Generalisation Challenge (OGC), the first open benchmark that jointly evaluates agents on *environment and partner generalisation* in cooperative multi-agent settings.

2. We release OvercookedUED – a JAX-accelerated, open-source extension of Overcooked-AI that supports unsupervised environment design (UED) via integration with state-of-the-art dual curriculum design (DCD) algorithms in `minimax` (Jiang et al., 2023) and JaxMARL (Rutherford et al., 2024b).

3. Through extensive experiments, we demonstrate that current DCD algorithms and scalable neural architectures – including recent state-of-the-art models – fail to generalise effectively across environments and partners, thus establishing OGC as a challenging new testbed for multi-agent cooperation.

## 2 Related Work

### 2.1 Partner Generalisation

Generalisation to novel partners has been studied under the ad-hoc teamwork (Stone et al., 2010) and zero-shot coordination (Hu et al., 2020) paradigms, both motivated by improving human-AI cooperation. A prominent benchmark in this space is Overcooked-AI (Carroll et al., 2019), where agents must jointly prepare and serve dishes. Many recent works use this environment to evaluate ad-hoc coordination capabilities (Strouse et al., 2021; Li et al., 2023b; Yan et al., 2023a; Liu et al., 2024; Tan et al., 2024).

In this setting, self-play often fails to produce agents that generalise to novel partners (Carroll et al., 2019). Consequently, researchers have turned to population-based methods that train diverse partner policies and learn best-response strategies in fixed environments (Zhao et al., 2023; Yu et al., 2023; Wang et al., 2024). However, these methods scale poorly, as training cost increases linearly with population size per environment Yan et al. (2023b).

In contrast, our setting trains agents across a large distribution of procedurally generated environments. Cooperation is evaluated on human-authored levels not seen during training – making population-based approaches infeasible and calling for learning strategies that operate effectively across novel partners and tasks.

### 2.2 Environment Generalisation

RL agents fail to generalise to new environments out-of-the-box (Zhang et al., 2018a) and instead require sufficiently diverse training levels to generalise well (Zhang et al., 2018b; Cobbe et al., 2019; 2020). One established approach to generate diverse training data is domain randomisation (DR; Jakobi, 1997). However, DR may produce uninformative samples (Khirodkar et al., 2018), which hinder learning (Dennis et al., 2020).

To improve sample quality, unsupervised environment design (UED) (Dennis et al., 2020) adaptively generates levels that match an agent's current capabilities. Prominent UED algorithms include PAIRED (Dennis et al., 2020), Prioritized Level Replay (PLR) (Jiang et al., 2021b), and ACCEL (Parker-Holder et al., 2022). These methods fall under the broader Dual Curriculum Design (DCD) framework (Jiang et al., 2021a), in which a generator and curator co-evolve to construct an adaptive training curriculum. While the development of DCD methods has been steady, they have mostly been explored in simple single-agent environments, e.g. in mazes (Dennis et al., 2020; Jiang et al., 2021a; Parker-Holder et al., 2022; Jiang et al., 2023; Li et al., 2023a; Beukman et al., 2024), bipedal walker (Wang et al., 2019; 2020; Parker-Holder et al., 2022) or car racing environments (Jiang et al., 2021a).

Multi-agent UED, in contrast, remains largely underexplored. Existing works are either closed source (Team et al., 2021; Bauer et al., 2023), do not address a (fully-)cooperative setting (Suarez et al., 2021; 2023; Samvelyan et al., 2023) or only feature multi-agent path-finding with no agent interaction (Rutherford

et al., 2024a). Moreover, the underlying design spaces are shallow – often involving only walls, agents, sparse control points or pregenerated levels without design control (Dennis et al., 2020; Parker-Holder et al., 2022; Samvelyan et al., 2023; Nikulin et al., 2023).

In contrast, OGC introduces a fully cooperative multi-agent UED environment with rich object interactions (e.g., pots, onions, plates) and complex spatial dependencies (see Figure 3). The difficulty of each task critically depends on object placement, making level design substantially more challenging. The OGC thus contributes the first open-source cooperative multi-agent UED environment in which agents are exposed to novel partners during evaluation.

## 2.3 Combining Partner and Environment Generalisation

While many benchmarks focus on either environment or partner generalisation (Lowe et al., 2017; Foerster et al., 2018; Hu et al., 2020), few evaluate both simultaneously. A recent Overcooked study explored cross-environment cooperation (Jha et al., 2025), but their approach relied on training agents across augmented variations of known test levels, therefore implicitly assuming access to the test distribution and significantly constraining the scope of generalisation.

In contrast, the OGC poses a strictly harder challenge: agents must learn to cooperate in **entirely novel environments and with unseen partners**, with **no prior exposure** to evaluation layouts or their structure. Training is conducted solely via procedurally generated levels using UED, without handcrafted augmentations or test-level tuning. The setting of (Jha et al., 2025) can be seen as addressing a reduced version of OGC where one assumes access to the testing layouts.

This decoupled setting reveals that current DCD algorithms struggle in high-complexity cooperative domains, and motivates a new class of approaches – **UED-ZSC methods** – that jointly tackle unsupervised environment design and zero-shot coordination. We propose OGC as both a benchmark and a testbed to support this emerging line of research, encouraging future methods that generalise across partners and environments in realistic, open-ended tasks.

## 3 Preliminaries

We formalise our cooperative multi-agent UED setting as a *decentralised under-specified partially observable Markov decision process* (Dec-UPOMDP) with shared rewards. A Dec-UPOMDP is defined as $\mathcal{M} = \langle \mathcal{N}, A, \Omega, \Theta, \mathcal{S}^{\mathcal{M}}, \mathcal{T}^{\mathcal{M}}, O^{\mathcal{M}}, \mathcal{R}^{\mathcal{M}}, \gamma \rangle$ in which $\mathcal{N}$ is the set of agents with cardinality $n$, $\Omega$ is a set of observations, and $\mathcal{S}^{\mathcal{M}}$ is the set of true states in the environment. Partial observations $o^i \in \Omega$ are obtained by agent $i \in \mathcal{N}$ using the observation function $O : \mathcal{S} \times \mathcal{N} \to \Omega$. Following Jiang et al. (2021a), a *level* $\mathcal{M}_\theta$ is defined as a fully-specified environment given some parameters $\theta \in \Theta$. In it, agents each pick an action $a_i \in A$ simultaneously to produce a joint action $\boldsymbol{a} = (a_1, \ldots, a_n)$ and observe a shared immediate reward $R(s, \boldsymbol{a})$. Then, the environment transitions to the next state according to a transition function $\mathcal{T} : \mathcal{S} \times \mathcal{A}^1 \times \ldots \times \mathcal{A}^n \times \Theta \to \Delta(\mathcal{S})$ where $\Delta(\mathcal{S})$ refers to the space of distributions over $\mathcal{S}$. $\gamma \in [0, 1)$ specifies the discount factor. Agents learn a policy $\pi$. The joint policy $\boldsymbol{\pi}$ together with the discounted return $R_t = \sum_{i=0}^{\infty} \gamma^i r_{t+1}$ induce a joint action value function $Q^{\boldsymbol{\pi}} = \mathbb{E}_{s_{t+1:\infty}, \boldsymbol{a}_{t+1:\infty}}[R_t | s_t, \boldsymbol{a}_t]$. Our formulation extends the Dec-POMDP framework (Oliehoek & Amato, 2016; Wu et al., 2021) by introducing $\Theta$ as a set of free environment parameters – making the model suitable for unsupervised environment design. This follows previous work (Dennis et al., 2020; Jiang et al., 2021a; Samvelyan et al., 2023), but differs from Samvelyan et al. (2023) in assuming shared rewards and a cooperative, rather than general-sum, structure. Within our Dec-UPOMDP, we perform UED to train a policy over a distribution of fully specified environments that enable optimal learning. This is facilitated by obtaining an *environment policy* $\Lambda$ (Dennis et al., 2020) that specifies a sequence of environment parameters $\Theta^T$ for the given policy that is to be trained. How $\Lambda$ is obtained depends on the DCD method. In OvercookedUED, $\Theta$ represents the possible positions of walls, pots, serving spots, agent starting locations, and onion and bowl piles adjusted by $\Lambda$ throughout training.

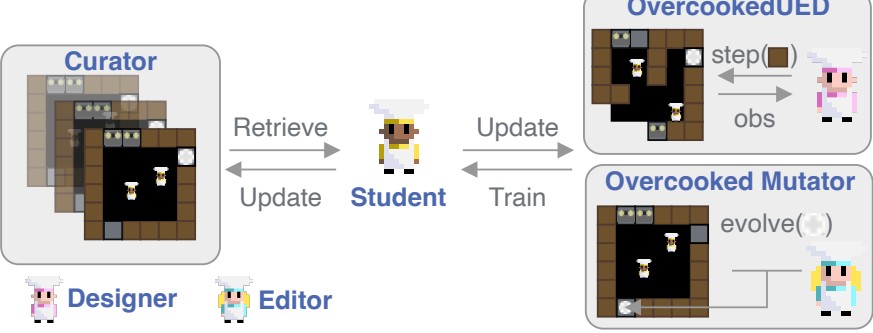

Figure 2: Overview of the OGC and how it is typically used in a DCD algorithm. The OGC supports teacher-based UED methods like PAIRED (Dennis et al., 2020) and edit-based methods like ACCEL (Parker-Holder et al., 2022) via mutator functions of existing layouts.

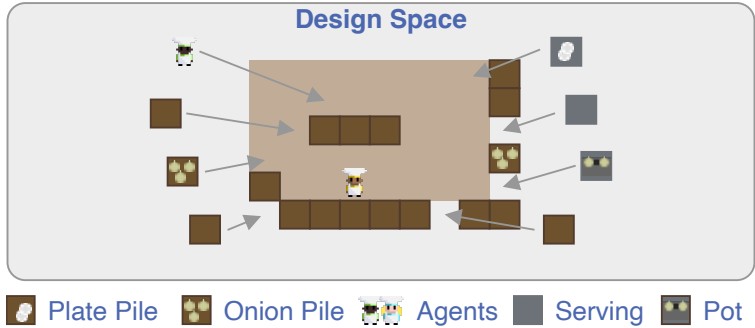

Figure 3: The OGC features a large design space in which many different elements have to be placed in relation to each other, creating challenging environments for both environment designers and agent training.

## 4 The Overcooked Generalisation Challenge

The Overcooked Generalisation Challenge is a new benchmark that allows us to evaluate agents on their ability to cooperate with unknown partners in previously unseen environments. Unlike existing benchmarks, the OGC combines unsupervised environment design with multi-agent coordination, introducing the first open-source UED testbed for cooperative RL covering both environment and partner generalisation. Built on Overcooked-AI Carroll et al. (2019), OGC integrates with DCD algorithms to support procedural training, layout mutation, and zero-shot evaluation across complex coordination tasks.

Overcooked-AI is one of the most popular environments for cooperative RL and ad-hoc teamwork. In the environment, two agents are tasked with cooking and delivering a soup together. Delivering a soup entails collecting and putting onions into a pot, letting it cook, collecting the soup with a plate and delivering it to a serving station, which results in a reward of 20.

Figure 2 shows how OGC interfaces with various DCD methods. It supports both teacher-based approaches (e.g., PAIRED (Dennis et al., 2020)) and edit-based methods (e.g., ACCEL (Parker-Holder et al., 2022)) using mutator functions that transform existing layouts. Figure 3 illustrates the complexity of layout design in Overcooked, where task difficulty depends critically on the spatial configuration of multiple interdependent objects and agents.

### 4.1 Environment and Layout Generation

OGC extends the JaxMARL implementation of Overcooked-AI (Rutherford et al., 2024b), which defines a discrete action space `left`, `right`, `up`, `down`, `interact`, `stay` and an observation space consisting of 26 binary masks of size $h \times w$, encoding the positions of agents, objects, and obstacles. To support large-scale

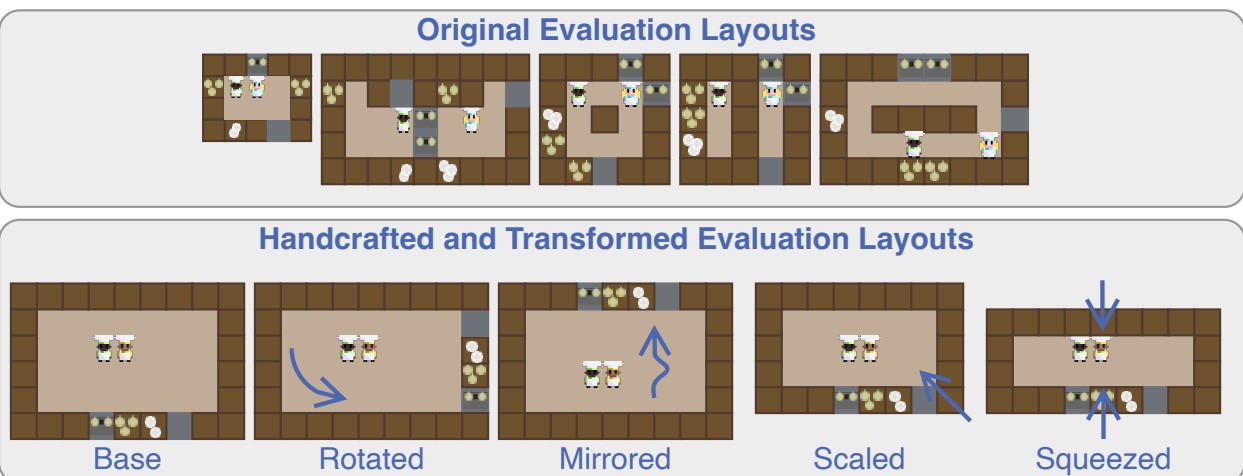

Figure 4: First, we propose to evaluate agents in self-play on the five original layouts and several layouts that are created from several symmetry classes to evaluate their ability to generalise. We combine a range of transformations shown in the bottom row to generate 28 additional layouts. Second, we evaluate coordination with novel partners in the original five.

training, we enable parallel rollouts across multiple layouts, requiring all layouts to be padded to a fixed maximum height $h$ and width $w$. This enables fast training and execution speeds across hundreds or thousands of environments using JAX.

## 4.2 Curriculum Learning in OGC

OGC exposes two core interfaces for DCD methods: OvercookedUED and the Overcooked Mutator.

OvercookedUED implements a teacher environment where a generator policy sequentially places objects onto a layout grid. At each step $t$, the teacher selects a grid cell and places one object from a fixed sequence (walls, agents, goals, ingredient piles, pots, bowls). If the target cell is already occupied, the object is placed randomly in a free cell of the same type. Placing two elements of the same type in the same location results in the second being ignored, enabling variable object counts per type – consistent with prior UED designs (Dennis et al., 2020). For UED methods that lack a teacher component (e.g., PLR), OvercookedUED also provides a random environment generator that follows the same structure as the teacher but samples object positions uniformly.

The Overcooked Mutator enables layout evolution for edit-based methods. It supports five operations: (1) toggling walls and free spaces, (2–5) moving goal, pot, plate, and onion pile positions. Agent start positions remain fixed. The number of mutations applied can be configured to control curriculum granularity.

All versions leave layout solvability unchecked, following the convention in prior UED work (Dennis et al., 2020; Jiang et al., 2023), and place responsibility for level quality on the DCD method.

## 4.3 Evaluation Protocol

We study three evaluation modes: Self-play, zero-shot and ad-hoc coordination.

We propose to test agents in self-play to evaluate how well they generalise to novel levels. Figure 4 illustrates our evaluation suite. We use the five original Overcooked-AI layouts (Carroll et al., 2019) and 32 layouts created via geometric transformations from a simple square base layout and randomly generated layouts to assess generalisation. The five original layouts enable comparisons to earlier work, and the transformed layouts expose whether the agent's behaviour is robust (or ideally invariant) to layout transformations that do not require a different strategy. We secondly propose to evaluate in an ad-hoc teamwork setting (Stone et al., 2010), we train populations for 24 agents for the original five layouts via: Fictitious Co-Play (FCP) (Strouse

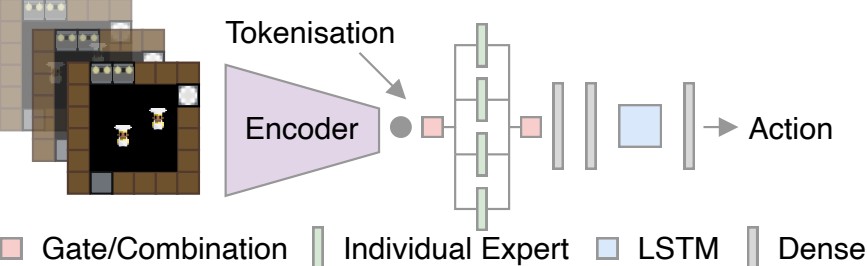

Figure 5: The SoftMoE-LSTM agents architecture used in this work. We employed the PerConv tokenisation technique introduced in Obando-Ceron et al. (2024).

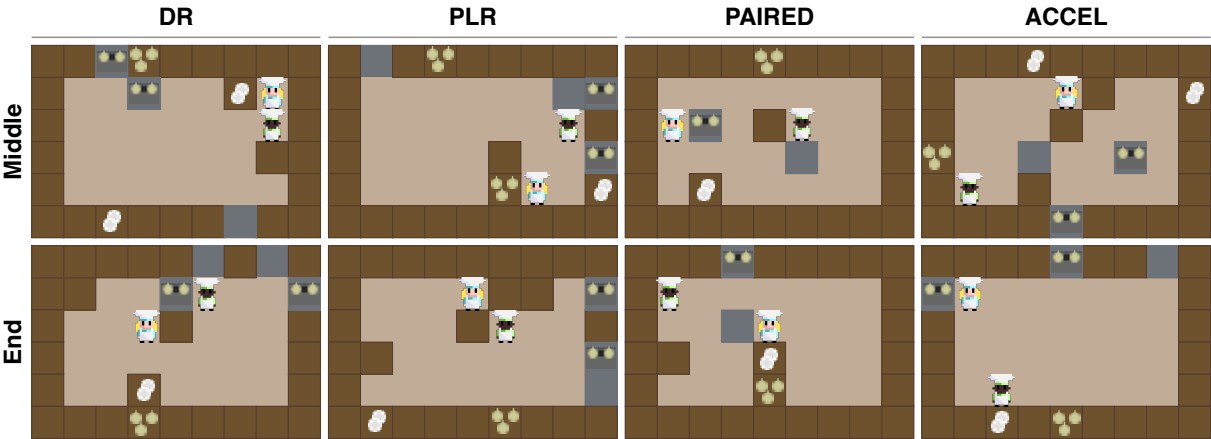

Figure 6: Sample levels generated by the different methods after $15,000$ (Middle) and $30,000$ (End) epochs. Even after considerable training, none of the methods can guarantee the generation of solvable layouts (Middle row, leftmost and rightmost).

et al., 2021) and Maximum Entropy Population Training (MEP) (Zhao et al., 2023). Each population includes low-, mid-, and high-skill checkpoints (10%, 50%, and 100% of achieved return) for diversity. MEP uses an entropy coefficient $\alpha = 0.01$. This setting evaluates the adaptability of trained agents to diverse, possibly brittle partners. Finally, we also evaluate in a zero-shot coordination setting (Hu et al., 2020) in which we test an agent's capability to adapt to partners that themselves were trained for the zero-shot coordination setting. Both ad-hoc teamwork and zero-shot coordination are evaluated on the original five.

We report two metrics. First, the mean episode reward and second, the solved rate: the proportion of episodes where at least two soups are delivered, distinguishing goal-directed behaviour from random actions. Finally, we also conduct a qualitative error analysis to examine failure modes across different levels.

Our primary evaluation metric based on episode return serves as a direct measure of both task efficiency and partner adaptability. Since rewards in Overcooked are only obtained by successfully delivering soups within a fixed time horizon, the return reflects how efficiently agents coordinate to complete tasks under temporal constraints. Additionally, measuring return in cross-play and ad-hoc teamwork settings isolates the agent's ability to adapt to novel partners without prior joint training, thus capturing adaptability in zero-shot cooperation.

## 5 Experiments

We conducted a series of experiments to establish performance baselines for partner and environment generalisation using state-of-the-art DCD methods and policy architectures. We first compare several policy architectures – CNN-LSTM, SoftMoE-LSTM, and S5-based models – on a fixed set of evaluation layouts to identify

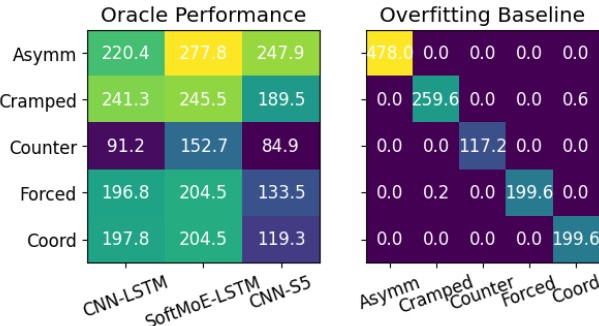

Figure 7: Return of policies in all evaluation layouts. **Left**: Results of policies trained across all five evaluation layouts to be used as an oracle. SoftMoE-LSTM shows the best performance. **Right**: Shows SoftMoE-LSTM agents trained only on one layout, which overfit.

a strong baseline. We then assess whether agents trained with different DCD methods can generalise to novel environments without prior exposure to the test distribution. This is followed by experiments to establish coordination capabilities with unseen partners by testing agents in ad-hoc team play against diverse policy populations (FCP, MEP), as well as in zero-shot coordination with agents trained under the same method but different random seeds. Finally, we analyse failure modes through a detailed error analysis, identifying structural layout characteristics that correlate with poor performance, such as object spacing and path complexity.

All agents were trained using MAPPO (Yu et al., 2022), a strong cooperative multi-agent baseline. For DCD, we tested diverse algorithms with distinct generation principles (Dennis et al., 2020; Jiang et al., 2023): domain randomisation (DR), priority-based replay (robust parallel PLR$^{\perp,\|}$), edit-based curriculum (parallel ACCEL$^{\|}$), and learned environment design (Pop. PAIRED). We excluded POET (Wang et al., 2019) in this analysis as it outputs specialists rather than generalists, which we require (Parker-Holder et al., 2022). Additionally, we excluded MAESTRO as it is based on prioritised fictitious self-play (Heinrich et al., 2015; Vinyals et al., 2019) that is not readily adaptable to the cooperative setting (Strouse et al., 2021). We chose these methods as they offer better theoretical guarantees (PLR$^{\perp}$ vs PLR), better runtime performance (ACCEL$^{\|}$ and PLR$^{\|}$), or because we found them to perform better empirically (Pop. PAIRED vs PAIRED). To ensure a fair comparison, we standardised the environment design space: each method placed between one and 15 walls (either randomly or through a learned teacher policy), along with one or two items per object category (pots, onions, serving points, etc.). Layouts were procedurally generated using either teacher actions (PAIRED) or stochastic editing (ACCEL, PLR), with training conducted across five seeds, 32 parallel environments for 30,000 training iterations ($\sim$ 400M steps). All architectural and training hyperparameters were selected via grid search and are detailed in Appendix A.4.

## 5.1 Experiment 1: Policies and Baselines

Before evaluating generalisation on OGC, we identified a strong agent architecture that can serve as a policy backbone across all experiments. Comparing network architectures allows us to: (1) understand how architecture affects generalisation, and (2) establish upper-bound *oracle* performance on evaluation layouts, which will serve as reference points throughout the paper. We explored the following three architectures (see Appendix A.5 for details): **CNN-LSTM** is a standard convolutional encoder followed by an LSTM that demonstrated strong performance in previous work (Yu et al., 2023). **SoftMoE-LSTM** is an enhanced architecture using a Soft Mixture-of-Experts (SoftMoE) module (Obando-Ceron et al., 2024) and PerConv tokenisation, replacing the final layer of the CNN-LSTM. We explore SoftMoE agents because of their strong parameter scaling properties (Obando-Ceron et al., 2024). **CNN-S5** is a CNN encoder paired with S5 layers (Smith et al., 2023) instead of LSTMs, inspired by structured state-space models (Gu et al., 2022) that showed strong performance in meta-RL (Lu et al., 2023). Each agent was trained on the five human-designed evaluation layouts (Cramped Room, Asymmetric Advantages, etc.) to assess whether the architecture could

Table 1: Mean episode reward for the different methods averaged over the respective testing layouts. The best result is shown in **bold**. We report aggregate statistics over five random seeds and the 95% confidence interval. We include oracles trained on the five testing layouts to establish an empirical upper bound.

| Method | CNN-LSTM | SoftMoE-LSTM | CNN-S5 |
|---|---|---|---|
| DR | $2.84 \pm 7.37$ | $4.21 \pm 9.52$ | $0.06 \pm 0.17$ |
| PLR | $0.18 \pm 0.05$ | $0.68 \pm 0.75$ | $0.13 \pm 0.14$ |
| PAIRED | $0.52 \pm 0.39$ | $\mathbf{13.12 \pm 11.46}$ | $0.02 \pm 0.44$ |
| ACCEL | $0.24 \pm 0.41$ | $0.61 \pm 0.55$ | $0.06 \pm 0.12$ |
| Oracle | $\mathbf{204.44 \pm 29.61}$ | $\mathbf{216.76 \pm 34.41}$ | $\mathbf{166.63 \pm 23.62}$ |

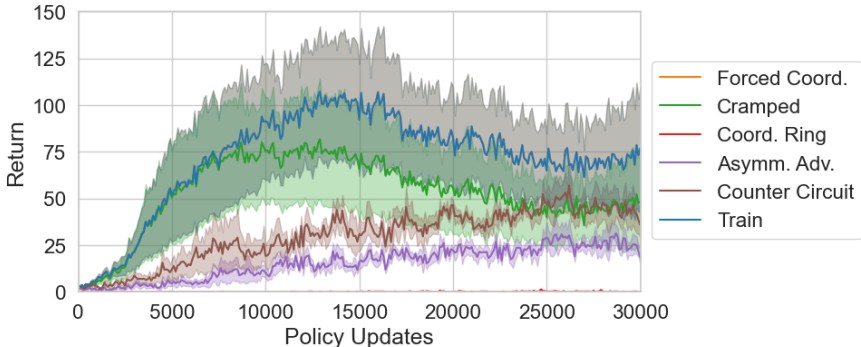

Figure 8: Returns over training for both training and evaluation layouts for our SoftMoE-LSTM-PAIRED policy. The policy has had some success in generalising, but its generalisation gap remains substantial.

fit the joint task. In all experiments that follow, we refer to these as **oracles** – they have access to test environments during training and thus represent an empirical upper bound without generalisation.

As shown in Figure 7 (left), all architectures are capable of fitting the evaluation layouts in self-play. **SoftMoE-LSTM achieves the highest returns** across the board (Table 1), with lower variance and better stability. CNN-S5 significantly underperforms, suggesting S5 layers may not suit cooperative RL in this setting. To confirm these models did not simply memorise layout-specific strategies, we trained a SoftMoE-LSTM agent on a single layout and evaluated it on all five. The steep performance drop suggests overfitting, underscoring the importance of multi-layout training.

**Key takeaway:** We find that *SoftMoE-LSTM generalises best among tested architectures*, and adopt it for all subsequent experiments. This result also suggests that mixture-of-expert routing may support generalisation in multi-object, sparse-reward environments like Overcooked.

## 5.2 Experiment 2: Generalisation to Novel Environments

We then tested whether agents trained with unsupervised environment design can generalise to unseen environments. Unlike recent work that trained on augmented variants of test levels (Jha et al., 2025), agents in the OGC are faced with entirely new test environments without prior exposure.

As can be seen in Table 1, despite using tuned implementations and scalable architectures, most methods fail to generalise, achieving near-zero returns on the evaluation layouts. Only Population PAIRED has limited success, significantly outperforming other methods ($p < 0.05$), with a mean solved rate of $14.6\% \pm 7.7$. All other methods barely go above 0% solved levels, suggesting that training on randomly generated or edited layouts is insufficient to prepare agents for the coordination structure of evaluation tasks. We check and find that roughly a quarter of randomly generated training layouts are unsolvable, and many others require long, inefficient action sequences. This further highlights why these methods struggle: not only is the evaluation set challenging, but producing *feasibly solvable* training levels is itself a key difficulty. The

Table 2: Performance on mirrored, rotated and squeezed levels as illustrated in Figure 4. The transformations are grouped such that for each category, the optimal strategy remains the same modulo mirroring and rotation. Large, medium and small are defined in terms of the available movement space. Squeezed and squeezed small define increasingly narrow spaces. The large confidence interval suggests that agents do not learn general strategies for Overcooked. We analyse SoftMoE-LSTM agents.

| Method | Large | Medium | Small | Squeezed | Squeezed Small | Average |
|--------|-------|--------|-------|----------|----------------|---------|
| DR | $18.1 \pm 30.1$ | $13.1 \pm 20.2$ | $10.4 \pm 15.8$ | $11.3 \pm 19.9$ | $10.3 \pm 17.4$ | $13.1 \pm 21.1$ |
| PLR | $0.2 \pm 0.4$ | $0.7 \pm 1.0$ | $1.0 \pm 0.6$ | $0.0 \pm 0.0$ | $0.1 \pm 0.1$ | $0.5 \pm 0.5$ |
| PAIRED | $16.2 \pm 17.6$ | $31.5 \pm 38.4$ | $25.4 \pm 28.7$ | $4.7 \pm 6.9$ | $8.4 \pm 8.9$ | $\mathbf{19.9 \pm 22.2}$ |
| ACCEL | $1.5 \pm 2.6$ | $1.4 \pm 0.6$ | $0.8 \pm 0.6$ | $0.2 \pm 0.4$ | $0.1 \pm 0.3$ | $0.9 \pm 0.8$ |

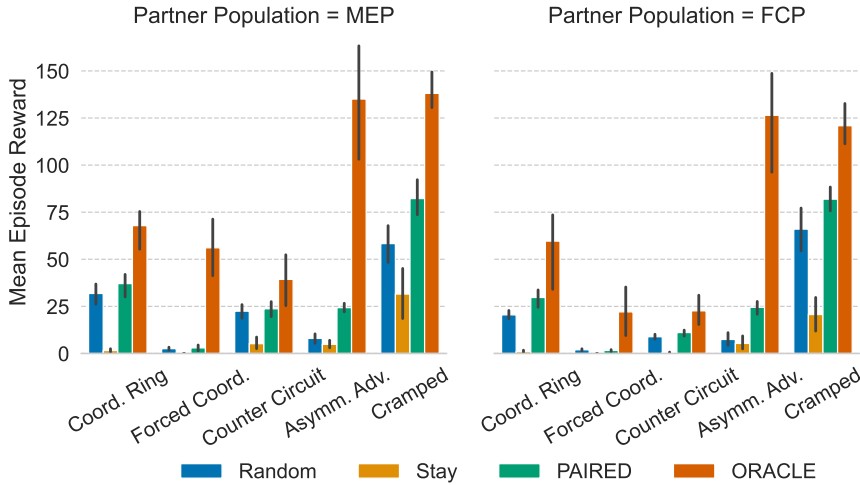

Figure 9: Ad-hoc team-play performance of SoftMoE-LSTM-PAIRED and other baselines with both an MEP and an FCP expert population. We measure the returns of multiple seeds across the five original layouts. Thinner lines on the bars indicate 95% confidence intervals around the means. While SoftMoE-LSTM-PAIRED outperforms simple baselines, it fails to reach oracle performance.

SoftMoE-LSTM-PAIRED policy only shows mediocre performance on *Asymmetric Advantages* and *Cramped Room* and completely fails to coordinate effectively in more complex layouts, such as *Counter Circuit* or *Forced Coordination*. The training curve in Figure 8 confirms this: While the SoftMoE agent converges on training layouts, its generalisation gap on evaluation levels remains substantial and persistent.

To probe this further, we evaluate agents on systematically transformed versions of base layouts – including mirrored, rotated, and squeezed variants – that preserve the underlying coordination task but alter spatial structure. Ideally, a general policy should perform consistently across such transformations. However, performance fluctuates widely (see Table 2), revealing that agents fail to learn spatially invariant coordination strategies. This suggests that current methods often overfit to superficial spatial patterns rather than acquiring abstract cooperation skills. However, even PAIRED exhibits high variance across layouts, highlighting its limited robustness.

**Key takeaway:** Even state-of-the-art DCD methods fail to generalise to unseen, complex multi-agent coordination tasks. This suggests that the OGC introduces a richer and more challenging design space than prior UED environments and reveals limitations in current level generation strategies – highlighting the need for stronger curriculum learning and generalisation-aware training algorithms.

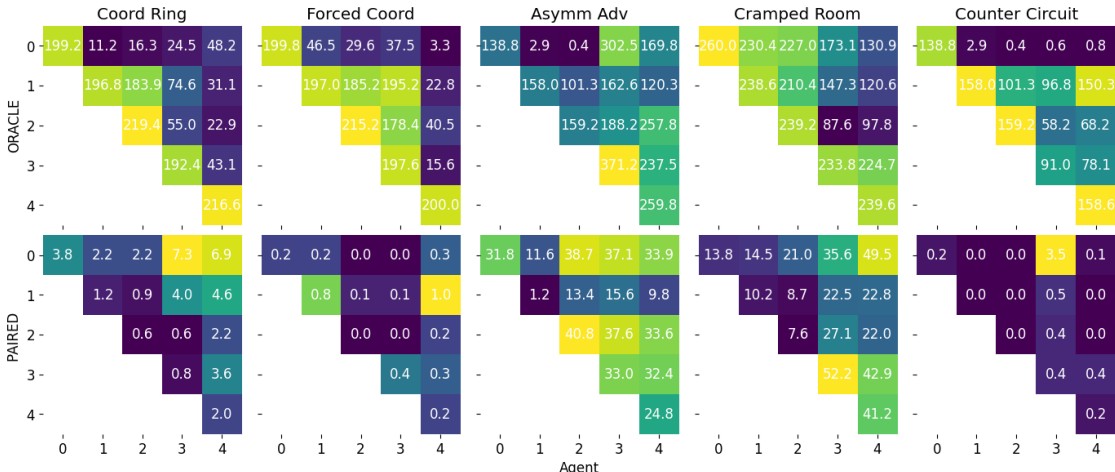

Figure 10: Zero-shot cooperation returns of the oracle and the SoftMoE-LSTM-PAIRED agents (trained with different seeds). Each square shows the performance of the row and column seeds. Self-play performance is thus displayed on the diagonal.

## 5.3 Experiment 3: Generalisation to Unknown Partners

To evaluate whether agents trained with DCD methods can coordinate with unknown partners in previously unseen environments, we investigated two settings: Ad-hoc teamplay and zero-shot coordination. We evaluated agents against diverse populations of pre-trained partners using two protocols: FCP (Strouse et al., 2021) and MEP (Zhao et al., 2023). Each population consisted of 24 agents with diverse skill levels and learning histories (see Appendix A.6.4).

Figure 9 shows the performance of the SoftMoE-LSTM-PAIRED agent compared to three baselines: a stationary partner (*stay*), a randomly acting agent (*random*), and an oracle trained on the evaluation layouts (see above). As can be seen from the figure, the SoftMoE agent consistently outperforms the baselines but fails to reach oracle performance. Notably, it often performs only slightly better than random coordination – a sign of poor robustness to novel partners. We hypothesise that this gap stems from the divergence between the training layouts (often open and simplified) and the evaluation layouts that have different cooperation demands. As shown in Figure 6, current DCD methods tend to converge toward minimal-complexity levels that facilitate early success but fail to expose agents to realistic partner dependencies.

We also assessed ZSC by evaluating whether the agent can coordinate with an independently trained copy of itself with a different random seed. This setting removes population diversity and isolates the agent's generalisation to novel weights and latent partner strategies. As shown in Figure 10, SoftMoE-LSTM-PAIRED performs poorly across the more complex evaluation layouts (*Coordination Ring*, *Forced Coordination*, *Counter Circuit*), and even underperforms its oracle counterpart. Interestingly, in the simpler layouts, the agent occasionally achieves higher returns in cross-play than in self-play – suggesting that some diversity across seeds may help mitigate overfitting.

**Key takeaway:** DCD-trained agents – despite some progress in environment generalisation – struggle to generalise to novel partners in unknown environments.

## 5.4 Experiment 4: Qualitative Failure Analysis

We perform a final experiment to investigate the agents' poor performance. To better understand which structures impede performance, we show cell visit patterns for the best and worst-performing layouts in Figure 11. While on many layouts, our PAIRED SoftMoE-LSTM agent reached good self-play performance (up to a maximum mean reward of 84.4; top row), it delivers few to no soups in others. Poorly performing layouts often feature narrow corridors or large object distances, suggesting that agents fail in environments requiring

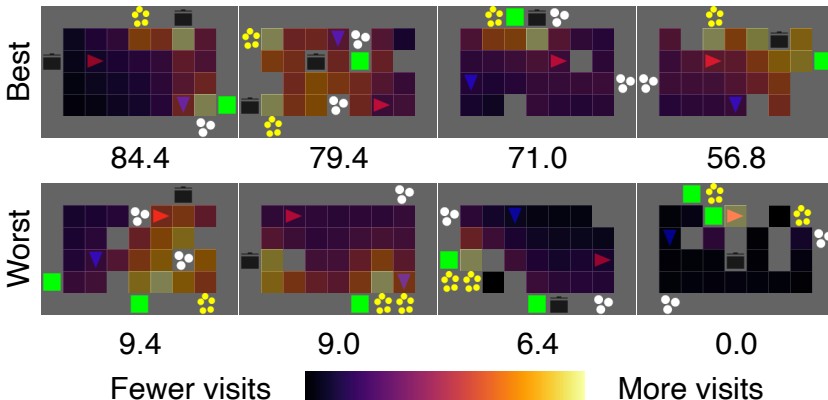

Figure 11: Sample levels in which our models perform best and worst. The number of visits to each grid cell is shown as a heatmap overlay, while the mean return is below. The worst layouts tend to feature narrow corridors or large distances between items.

fine-grained coordination and collision avoidance. We previously identified a key failure mode: (1) agents fail to learn spatially invariant coordination strategies. This experiment reveals a second: (2) performance degrades significantly in more complex spatial layouts, particularly those that demand structured movement and proximity-based interaction. These insights underline the difficulty of OGC and the need for curriculum strategies that better expose agents to high-complexity, high-coordination scenarios during training.

## 6 Discussion

Our results suggest that generalising to both novel partners and novel environments remains a fundamental challenge in cooperative reinforcement learning. While e.g. Jha et al. (2025) explored cross-environment play, a preliminary analysis showed that their agents also do not generalise to entirely novel layouts/partners. To show this, we retrained their agents and tested them on the 32 evaluation layouts discussed in Experiment 2. When evaluated on the 32 transformed layouts from Experiment 2, their method only achieved an average return of $0.46 \pm 2.6$, performing worse than the proposed UED-based approaches (e.g., PAIRED with SoftMoE-LSTM). These agents appear to learn brittle strategies tied to the five original Overcooked layouts, and are unable to adapt to structural variations or unfamiliar partners.

Our findings also have direct implications for future research on UED and DCD: While previous studies, such as Jiang et al. (2021a), found that $PLR^{\perp}$ outperformed other DCD methods in navigation-based tasks, we show that this result does not generalise to more complex, multi-agent cooperative environments. In our experiments, PAIRED consistently outperformed PLR and ACCEL. We attribute this to the increased design space complexity offered by the OGC: The environment contains multiple object types, coordination bottlenecks, and sparse rewards, all of which demand deliberate environment construction. In simpler domains like mazes or locomotion tasks, randomly generated or lightly curated curricula may suffice. In OvercookedAI, however, this approach fails to expose agents to the kinds of structured coordination tasks they must solve at test time.

This leads to three key conclusions: **First**, *DCD methods must scale with environment complexity.* Benchmarks that rely on narrow design spaces (e.g., only walls in a maze) are insufficient for evaluating the capabilities of curriculum-learning algorithms. The OGC reveals that without principled curriculum generation, agents may never encounter useful learning signals. **Second**, *Current DCD methods do not scale natively to realistic cooperative tasks.* Even with tuned architectures and training regimes, we observed poor generalisation across environments and partners. This highlights the importance of new methods that integrate environmental and partner generalisation. **Third**, the *OGC provides a critical testbed for advancing this next generation of methods.* By supporting both axes of generalisation, the OGC offers a foundation for future research into UED-ZSC methods capable of producing robust, general-purpose cooperative agents that perform well in open-ended multi-agent settings.

## 7 Limitations & Future Work

Despite its advantages, we also identified two limitations of the OGC: First, to support parallel training and JAX-based acceleration, we constrain all layouts to a fixed maximum height and width. While we included a partial observation that can theoretically be computed independently of size, similar to the vector-based observation used for behaviour cloning agents in (Carroll et al., 2019), batching across layouts in OvercookedUED still requires the layouts to be scaled to the same height and width. Future work could explore layout representations that scale more naturally, such as graph-structured inputs or object-centric embeddings, to remove these spatial limitations.

Second, while OGC evaluates coordination under environmental and partner variation, it does not explicitly test agents' ability to reason about their partners' beliefs, intentions, or mental models. Such capabilities – often studied under theory-of-mind or agent modelling frameworks (Rabinowitz et al., 2018; Bard et al., 2020; Gandhi et al., 2021; Bara et al., 2023; Bortoletto et al., 2024b;a) – are likely to be important for achieving robust zero-shot human-AI collaboration. Future work could explore reasoning about other agents in previously unexplored environments.

Finally, while our experiments focus on DCD methods, OGC is compatible with a wide range of generalisation approaches. Future work could for instance explore few-shot adaption to partners and environments (Bauer et al., 2023) or explore representation learning approaches for better generalisation. We hope that the benchmark we provide will inspire and support further work in these areas.

## 8 Conclusion

We introduced the Overcooked Generalisation Challenge (OGC), the first open-source benchmark for evaluating cooperative multi-agent reinforcement learning (MARL) agents on both environment and partner generalisation. Built on Overcooked-AI and integrated with dual curriculum design (DCD) methods, OGC enables procedural training and rigorous testing in complex, multi-object environments. Compared to prior UED benchmarks, OGC presents a significantly larger and more structured design space, exposing key limitations in existing environment generation and coordination strategies. Through extensive experiments, we demonstrated that even state-of-the-art DCD algorithms struggle to train agents that generalise across layouts and partners. These findings position OGC as a diagnostic tool for probing the frontiers of generalisable cooperation. Beyond DCD research, OGC also provides infrastructure for evaluating human-AI interaction through ad-hoc teamwork and zero-shot coordination. We hope that OGC will catalyse the development of new learning methods – what we denote UED-ZSC algorithms – that jointly address the challenges of task and partner diversity in open-ended multi-agent settings.

### Broader Impact Statement

This work introduces a benchmark for studying generalisation in cooperative multi-agent learning. While fundamental, it may inform future systems that support human-AI collaboration in domains such as assistive robotics or simulation-based training. Poor generalisation in such settings could lead to coordination breakdowns if deployed without safeguards.

### Acknowledgments

The authors thank the International Max Planck Research School for Intelligent Systems (IMPRS-IS) for supporting C. Ruhdorfer. A. Penzkofer was funded by the Deutsche Forschungsgemeinschaft (DFG, German Research Foundation) under Germany's Excellence Strategy – EXC 2075 – 390740016.

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

# A    Appendix

## A.1    Accessibility of the Benchmark

We make our challenge available under the Apache License 2.0 via a code repository: `https://git.hcics.simtech.uni-stuttgart.de/public-projects/OGC`. Our environment is built on top of the existing `minimax` project (accessible under Apache License 2.0 via `https://github.com/facebookresearch/minimax`) and is thus accessible to researchers who are already familiar with the project. `minimax` is extensively documented, fast, and supports multi-device training. For all details, including a full description of the advantages of `minimax`, we kindly refer the reader to the accompanying publication (Jiang et al., 2023). Our Overcooked adaption is extended from the one in JaxMARL also accessible under Apache License 2.0 via `https://github.com/FLAIROx/JaxMARL`. Our code includes extensive documentation and examples of how it may be used. Additionally, our code is written in a modular fashion and other multi-agent environments can be integrated with the runners.

## A.2    Infrastructure & Tools

We ran our experiments on a server running Ubuntu 22.04, equipped with NVIDIA Tesla V100-SXM2 GPUs with 32GB of memory and Intel Xeon Platinum 8260 CPUs. All training runs are executed on a single GPU only. We trained our models using Jax (Bradbury et al., 2018) and Flax (Heek et al., 2023) with `1`, `2`, `3`, `4` and `5` as random seed for training DCD methods and `1` to `8` as random seeds for the populations. Training the DCD methods usually finishes in under 24 hours, only SoftMoE and PAIRED-based methods take longer. SoftMoE-based policies often take an extra 50% wall-clock time to train. Noticeable is also that our S5 implementation is the fastest, usually needing roughly 30% less time. Both are compared to the default architectures' training time. In the longest case, the combination of a SoftMoE-LSTM policy trained with PAIRED takes about 80 hours to complete training. Our benchmark should be runnable on any system that features a single CUDA-compatible GPU. Although in our experience our experiments will require 32GB VRAM to run.

## A.3    Extended Related Work

We present an overview over how our environment compares to other UED environments in Table 3.

## A.4    Hyperparameters

We overview all hyperparameters for training in Table 4 and provide details on the hyperparameter search used in Table 5. This search was conducted on smaller single layout runs to determine reasonable values as complete runs would have been computationally infeasible. Furthermore we show the hyperparameters for each DCD method separately: DR hyperparameters in Table 6, PLR hyperparameters in Table 7, ACCEL hyperparameters in Table 8, and PAIRED hyperparameters in Table 9. DR hyperparameters govern how Overcooked levels are generated randomly and apply to all other processes in which a random level is sampled, for instance, in PLR, in which case the same hyperparameters apply.

In the experiment displayed in Figure 7 we show how policies behave when trained on all versus on only one layout and then decide on the SoftMoE architecture for our agents. Since these are considerably easier

Table 3: Overview of benchmarks for unsupervised environment design and procedurally generated environments. Closed-source benchmarks are marked in gray – these cannot be evaluated on by the research community.

| Name | Multi-agent | Zero-shot coop. | GPU accel-erated | Open Source | Partial obs. | Img. obs. |
|---|---|---|---|---|---|---|
| XLand (Team et al., 2021; Bauer et al., 2023) | ✓ | ✓ | - | ✓ | ? | ✓ |
| LaserTag (Samvelyan et al., 2023) | ✓ | - | - | - | ✓ | ✓ |
| MultiCarRacing (Samvelyan et al., 2023) | ✓ | - | - | - | ✓ | ✓ |
| CoinRun (Cobbe et al., 2019) | - | - | - | ✓ | ✓ | ✓ |
| ProcGen (Cobbe et al., 2020) | - | - | - | ✓ | ✓ | ✓ |
| 2D Mazes (Cobbe et al., 2019; Dennis et al., 2020) | - | - | - | ✓ | ✓ | ✓ |
| CarRacing (Jiang et al., 2021a) | - | - | - | ✓ | ✓ | ✓ |
| Bipedal Walker (Wang et al., 2019) | - | - | - | ✓ | ✓ | - |
| AMaze (Jiang et al., 2023) | - | - | ✓ | ✓ | ✓ | ✓ |
| XLand-MiniGrid (Nikulin et al., 2023) | - | - | ✓ | ✓ | ✓ | ✓ |
| Craftax (Matthews et al., 2024) | - | ✓ | ✓ | ✓ | - | ✓ |
| JaxNav (Rutherford et al., 2024a) | ✓ | - | ✓ | ✓ | ✓ | - |
| **OvercookedUED (ours)** | ✓ | ✓ | ✓ | ✓ | ✓ | ✓ |

Table 4: Hyperparamters of the learning process.

| Description | Value |
|---|---|
| Optimizer | Adam (Kingma & Ba, 2015) |
| Adam $\beta_1$ | 0.9 |
| Adam $\beta_2$ | 0.999 |
| Adam $\epsilon$ | $1 \cdot 10^{-5}$ |
| Learning Rate $\eta$ | $3 \cdot 10^{-4}$ |
| Learning Rate Annealing | - |
| Max Grad Norm | 0.5 |
| Discount Rate $\gamma$ | 0.999 |
| GAE $\lambda$ | 0.98 |
| Entropy Coefficient | 0.01 |
| Value Loss Coefficient | 0.5 |
| # PPO Epochs | 8 |
| # PPO Minibatches | 4 |
| # PPO Steps | 400 |
| PPO Value Loss | Clipped |
| PPO Value Loss Clip Value | 0.2 |
| Reward Shaping | Yes (linearly decreased over training) |

problems we train these models on fewer environment steps in total. We train the overfitting baseline for roughly 1/30 (12, 800, 000 steps in the environment) of the experience of all UED methods and only use 1, 000 outer loops. For the Oracle baseline, we use 1/2 of the experience (200, 000, 000 steps in the environment) and only 5, 000 outer loops. Both are trained until convergence, and to speed up training, we deploy 100 environment simulators. Recall that the UED methods use 30, 000 training loops and 400, 000, 000 steps in the environment. Notably, in the case of population PAIRED these steps apply per student and do not include any additional steps taken in the teacher environment.

Table 5: Values used for a grid search over hyperparameters governing the learning process. Finally used values appear in **bold**.

| Description | Value |
|---|---|
| Learning Rate $\eta$ | $[1 \cdot 10^{-4}, \mathbf{3 \cdot 10^{-4}}, 5 \cdot 10^{-4}, 1 \cdot 10^{-3}]$ |
| Entropy Coefficient | $[\mathbf{0.01}\ 0.1]$ |
| # PPO Steps | $[256, \mathbf{400}]$ |
| # Hidden Layers | $[2, \mathbf{3}, 4]$ |
| Reward Shaping Annealing Steps | $[0, 2500000, 5000000, \mathbf{until\ end}]$ |

Table 6: DR hyperparameters.

| Description | Value |
|---|---|
| $n$ walls to place | Sampled between $0 - 15$ |
| $n$ onion piles to place | Sampled uniformly between $1 - 2$ |
| $n$ plate piles to place | Sampled uniformly between $1 - 2$ |
| $n$ pots to place | Sampled uniformly between $1 - 2$ |
| $n$ goals to place | Sampled uniformly between $1 - 2$ |

### A.5   Neural Network Architectures

This work employs an actor-critic architecture using a separate actor and critic in which the critic is centralised for training via MAPPO (Yu et al., 2022). For the actor, the observations are of shape $h \times w \times 26$, while for the centralised critic, we concatenate the observations along the last axis to form a centralised observation, i.e. the centralised observation has shape $h \times w \times 52$ following prior work (Yu et al., 2023).

All our networks feature a convolutional encoder $f_c$. This encoder always features three 2D convolutions of 32, 64 and 32 channels with kernel size $3 \times 3$ each and pads the input with zeros. Our default activation function is ReLU (Fukushima, 1975; Nair & Hinton, 2010) which we apply after every convolutional block. We feed the output of $f_c$ to a feed-forward neural network $f_e$ with three layers with 64 neurons, ReLU and LayerNorm (Ba et al., 2016) applied each. $f_e$ takes the flattened representation produced by $f_c$ and produces an embedding $e \in \mathbb{R}^{b \times t \times 64}$ that we feed into a recurrent neural network (either LSTM (Hochreiter & Schmidhuber, 1997) or S5 (Smith et al., 2023)) to aggregate information along the temporal axis. We use this resulting embedding $e_t \in \mathbb{R}^{b \times 64}$ to produce action logits $l \in \mathbb{R}^{b \times 6}$ to parameterise a categorical distribution in the actor-network or directly produce a value $v \in \mathbb{R}^{b \times 1}$ in the critic network using a final projection layer. This architecture is inspired by previous work on Overcooked-AI, specifically (Yu et al., 2023), see Figure 12 for an overview. We also test the use of a S5 layer (Smith et al., 2023) in which case we use 2 S5 blocks, 2 S5 layers, use LayerNorm before the SSM block and the activation function described in the original work, i.e. $a(x) = \text{GELU}(x) \odot \sigma(W * \text{GELU}(x))$. In the case of the SoftMoE architecture, we follow the same approach as in (Obando-Ceron et al., 2024) and replace the penultimate layer with a SoftMoE layer. As in their work we use the PerConv tokenisation technique, i.e. given input $x \in \mathbb{N}^{h \times w \times 26}$ we take the output $y \in \mathbb{R}^{h \times w \times 32}$ of $f_c$ and construct $h \times w$ tokens with dimension $d = 32$ that we then feed into the SoftMoE layer. We always use 32 slots and 4 experts for this layer, see (Obando-Ceron et al., 2024) for details on this layer. The resulting embedding is then passed into the two remaining linear layers before being also passed to RNN and used to produce an action or value, equivalent to the description above, also compare Figure 5.

Lastly, we describe our networks in terms of parameter count in Table 10.

Table 7: PLR specific hyperparameters in addition to the DR hyperparameters.

| Description | Value |
|---|---|
| UED Score | MaxMC (Jiang et al., 2021a) |
| PLR replay probability $\rho$ | 0.5 |
| PLR buffer size | 4,000 |
| PLR staleness coefficient | 0.3 |
| PLR temperature | 0.1 |
| PLR score ranks | Yes |
| PLR minimum fill ratio | 0.5 |
| PLR$^{\perp}$ | Yes |
| PLR$^{\parallel}$ | Yes |
| PLR force unique level | Yes |

Table 8: ACCEL hyperparameters in addition to the DR hyperparameters.

| Description | Value |
|---|---|
| UED Score | MaxMC (Jiang et al., 2021a) |
| PLR replay probability $\rho$ | 0.8 |
| PLR buffer size | 4,000 |
| PLR staleness coefficient | 0.3 |
| PLR temperature | 0.1 |
| PLR score ranks | Yes |
| PLR minimum fill ratio | 0.5 |
| PLR$^{\perp}$ | Yes |
| PLR$^{\parallel}$ | Yes |
| PLR force unique level | Yes |
| ACCEL Mutation | Overcooked Mutator |
| ACCEL $n$ mutations | 20 |
| ACCEL subsample size | 4 |

### A.6    Additional Analysis

#### A.6.1    Implementation Details

The OGC is implemented in Jax (Bradbury et al., 2018) and integrated into `minimax` (Jiang et al., 2023). As such, it can be tested with all available DCD algorithms present in `minimax`. To achieve this we extend `minimax` with runners, replay buffers etc. that are compatible with multiple agents. Building on an established library eliminates sources of error and presents users of the challenge with a familiar experience. We present the steps-per-seconds (SPS) on our setup given varying degrees of parallelism in Table 11 and compare it to the GPU-accelerated maze environment `minimax` includes AMaze. Given sufficiently large numbers of parallel environments, OGC can be run at hundreds of thousands of SPS. While less than AMaze, the OGC is a more fully-featured environment in which multiple agents take steps and interact.

#### A.6.2    Performance Across Levels

To accompany the overall performance measured by reward in the main paper in Table 1 we also measure the mean solved rate in Table 12.

#### A.6.3    Performance on Individual Levels

We list the performance of every individual method on every single layout in Table 13. Most notable is that some layouts are harder to learn than others. Our agents especially seem to struggle with layouts requiring

Table 9: PAIRED hyperparameters. All PPO hyperparameters are the same between the student and the teacher. The `minimax` implementation follows to original one in (Dennis et al., 2020) and we stick to it too.

| Description | Value |
|---|---|
| $n$ walls to place | Sampled between $0 - 15$ |
| $n$ students | 2 |
| UED Score | Relative regret (Dennis et al., 2020) |
| UED first wall sets budget | Yes |
| UED noise dim | 50 |
| PAIRED Creator | OvercookedUED |

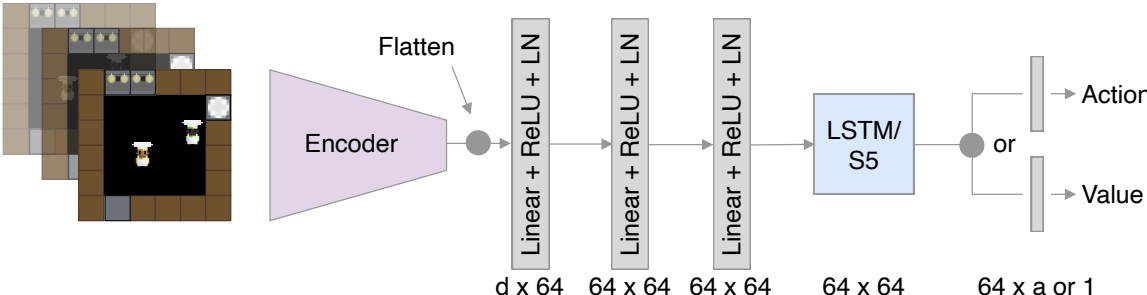

Figure 12: Default architecture featuring a convolutional encoder and an RNN.

more complex forms of interaction, i.e. Coordination Ring, Counter Circuit and Forced Coordination. Forced Coordination especially seems difficult to solve as no run achieves noticeable performance on it. This might be due to the specific features of the layout, i.e. agents have access to several objects and need to hand them over the counter to produce any result.

### A.6.4  Population Training Details

Both populations were trained over 8 random seeds. As architecture, we used a simple CNN encoder without RNN as in prior work Zhao et al. (2023); Yu et al. (2023). To give an intuition into the performance of the members of the population, we present the training curves over all 8 seeds of training an FCP population in Figure 13. MEP was trained with the same architecture, with the same amount of experience per agent and achieved practically identical results. As in prior work (Zhao et al., 2023) we set the population entropy coefficient during training to $\alpha = 0.01$.

### A.6.5  Detailed Results with Populations

We present detailed zero-shot cooperation results per layout in Table 14 and 15. As indicated through the averaged performance discussed in the main text, we also find that PAIRED performs best on four of the five individual layouts in terms of zero-shot cooperation.

### A.7  Training Curves and Evaluation

In Figure 15, Figure 16 and Figure 17 we show the returns of our agent during training in seen training levels, as well as the five unseen evaluation levels. The results for the SoftMoE architecture are displayed in Figure 15, the results for the S5 in Figure 16 and the results for the CNN-LSTM in Figure 17. Interestingly, while (SoftMoE) PAIRED performs the best in our evaluations it does not reach the highest training returns, instead it achieves the highest training return, while keeping the generalisation gap small.

Table 10: Number of trainable parameters in each model.

|                 | CNN-LSTM | SoftMoE-LSTM | CNN-S5  |
|-----------------|----------|--------------|---------|
| Parameter Count | 197,254  | 316,102      | 193,670 |

Table 11: Average steps-per-second for different numbers of parallel environments measured by taking 1,000 steps with randomly sampled actions to show how our adapted Overcooked environment performs compared to a simpler single-agent UED environment.

| # Parallel Envs | 1   | 32    | 256    | 1024    | 4096      | 16384     |
|-----------------|-----|-------|--------|---------|-----------|-----------|
| AMaze           | 264 | 8,141 | 67,282 | 264,142 | 1,058,306 | 3,321,678 |
| OvercookedUED   | 151 | 4,921 | 40,011 | 156,696 | 631,836   | 2,017,526 |

Table 12: Mean Solve rate averaged over the respective testing layouts. The best result is shown in **bold**. We report aggregate statistics over five random seeds. As a baseline we include an Oracle version for all architectures, which was trained on the five testing layouts directly.

| Method                  | CNN-LSTM        | SoftMoE-LSTM             | CNN-S5          |
|-------------------------|-----------------|--------------------------|-----------------|
| DR                      | $4.81 \pm 13.3\%$ | $3.78 \pm 9.8\%$         | $0.00 \pm 0.0\%$ |
| PLR$^{\perp,\parallel}$ | $0.03 \pm 0.0\%$  | $3.68 \pm 9.9\%$         | $0.00 \pm 0.0\%$ |
| Pop. PAIRED             | $0.08 \pm 0.2\%$  | $\mathbf{14.72 \pm 12.2\%}$ | $0.00 \pm 0.0\%$ |
| ACCEL$^{\parallel}$     | $0.00 \pm 0.0\%$  | $0.04 \pm 0.1\%$         | $0.00 \pm 0.0\%$ |
| Oracle                  | $97.2 \pm 7.7\%$  | $99.7 \pm 0.6\%$         | $97.2 \pm 2.8\%$ |

Table 13: Performance on all evaluation layouts. We show the mean episode reward **R** and the mean episode solved rate **SR**. The overall best result per layout is presented in **bold**, excluding oracle results.

| Layout | Method | CNN-LSTM | | SoftMoE-LSTM | | CNN-S5 | |
|---|---|---|---|---|---|---|---|
| | | **R** | **SR** | **R** | **SR** | **R** | **SR** |
| Cramped | DR | 0.02 | 0.0% | 0.01 | 0.0% | 0.00 | 0.0% |
| | PLR$^{\perp,\parallel}$ | 0.00 | 0.0% | 0.16 | 2.1% | 0.01 | 0.0% |
| | Pop. PAIRED | 0.78 | 0.0% | **8.92** | **9.0%** | 0.00 | 0.0% |
| | ACCEL$^{\parallel}$ | 1.16 | 2.7% | 1.52 | 3.4% | 0.28 | 2.1% |
| | Oracle | 197.56 | 100.0% | 201.56 | 100.0% | 129.64 | 94.0% |
| Coord | DR | 4.98 | 0.0% | 13.2 | 0.0% | 0.00 | 0.0% |
| | PLR$^{\perp,\parallel}$ | 0.09 | 0.0% | 0.44 | 6.6% | 0.06 | 0.0% |
| | Pop. PAIRED | 0.4 | 0.0% | **28.76** | **18.0%** | 0.1 | 0.0% |
| | ACCEL$^{\parallel}$ | 0.00 | 0.0% | 0.28 | 0.0% | 0.00 | 0.0% |
| | Oracle | 278.56 | 100.0% | 293.00 | 100.0% | 252.6 | 100.0% |
| Forced | DR | 1.78 | 0.0% | **3.48** | 0.0% | 0.00 | 0.0% |
| | PLR$^{\perp,\parallel}$ | 0.00 | 0.0% | 0.50 | 0.5% | 0.00 | 0.0% |
| | Pop. PAIRED | 0.04 | 0.0% | 2.64 | **6.0%** | 0.00 | 0.0% |
| | ACCEL$^{\parallel}$ | 0.00 | 0.0% | 0.04 | 0.0 % | 0.00 | 0.0 % |
| | Oracle | 194.12 | 100.0% | 204.96 | 100.0% | 146.69 | 96.0% |
| Asymm | DR | 7.32 | 0.1% | 3.38 | 4.4% | 0.32 | 0.0% |
| | PLR$^{\perp,\parallel}$ | 0.82 | 0.0% | 3.29 | 11.2% | 0.62 | 0.1% |
| | Pop. PAIRED | 1.68 | 0.6% | **33.42** | **40.4%** | 0.10 | 0.0% |
| | ACCEL$^{\parallel}$ | 0.04 | 0.0% | 1.20 | 0.0 % | 0.00 | 0.0 % |
| | Oracle | 238.68 | 100.0% | 242.40 | 98.4% | 206.36 | 99.7% |
| Counter | DR | 0.11 | 0.0% | 0.96 | 0.0% | 0.00 | 0.0% |
| | PLR$^{\perp,\parallel}$ | 0.00 | 0.0% | 0.00 | 0.0% | 0.00 | 0.0% |
| | Pop. PAIRED | 0.00 | 0.0% | **1.12** | 0.0% | 0.00 | 0.0% |
| | ACCEL$^{\parallel}$ | 0.00 | 0.0% | 0.00 | 0.0% | 0.00 | 0.0% |
| | Oracle | 113.28 | 86.0% | 142.00 | 100.0% | 95.60 | 96.0% |

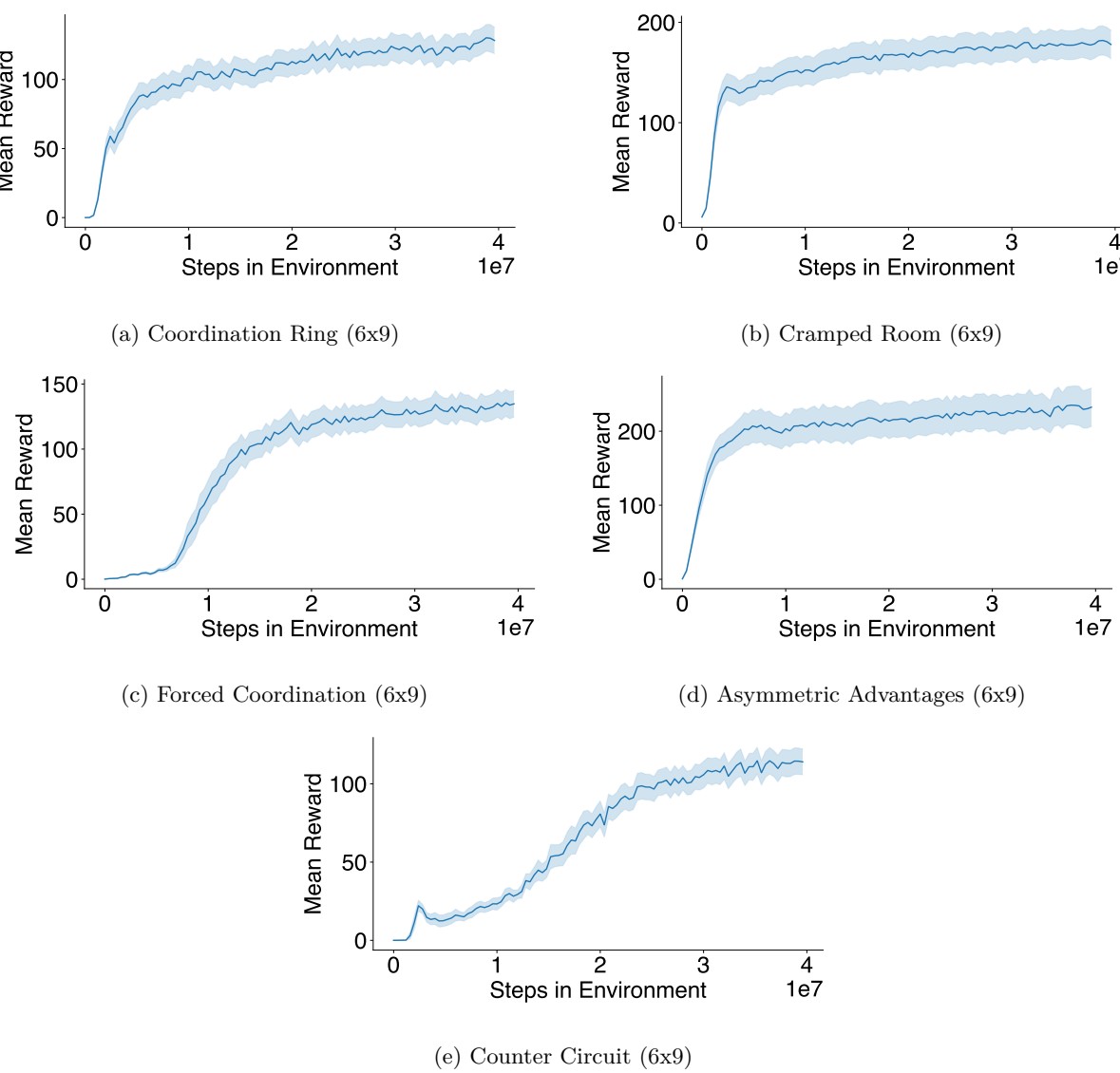

Figure 13: Runs used for the FCP evaluation populations with random seeds $1-8$ for the OGC with bands reporting standard error $\sigma/\sqrt{n}$. Layouts were padded to a total size of 6 x 9 to be compatible with the policies trained via DCD.

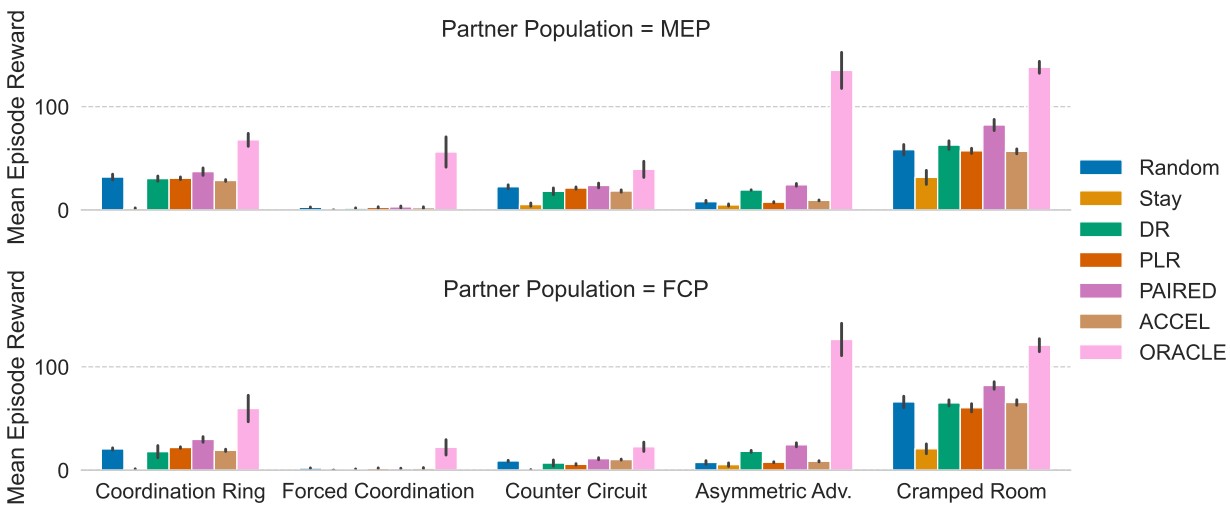

Figure 14: Ad-hoc teamwork results of the SoftMoE-LSTM policy paired with an FCP and MEP population trained on the respective layout. Error bars show standard error.

Table 14: Zero-shot results using SoftMoE-LSTM policies playing with an FCP and MEP population of experts trained on the respective layout exclusively. We report the *mean episode reward* and standard deviation. The best result per layout is put in **bold**.

| Method | Asymm | Counter | Cramped | Forced | Coord |
|---|---|---|---|---|---|
| **FCP** | | | | | |
| Random | $7.43 \pm 12.19$ | $8.89 \pm 4.65$ | $66.02 \pm 38.28$ | $1.95 \pm 1.92$ | $20.49 \pm 7.82$ |
| Stay | $5.32 \pm 12.07$ | $0.38 \pm 1.11$ | $20.67 \pm 33.05$ | $0.00 \pm 0.00$ | $0.95 \pm 2.73$ |
| Oracle | $126.44 \pm 27.13$ | $22.63 \pm 7.82$ | $120.9 \pm 10.86$ | $22.08 \pm 12.89$ | $59.64 \pm 22.17$ |
| DR | $19.47 \pm 8.79$ | $9.02 \pm 4.67$ | $59.62 \pm 10.49$ | $1.32 \pm 0.60$ | $21.63 \pm 11.01$ |
| PLR$^{\perp, \parallel}$ | $8.75 \pm 1.99$ | $7.00 \pm 2.13$ | $60.85 \pm 5.03$ | $1.78 \pm 0.75$ | $20.28 \pm 2.41$ |
| Pop. PAIRED | $\mathbf{45.71 \pm 14.65}$ | $\mathbf{11.72 \pm 2.49}$ | $\mathbf{72.2 \pm 13.86}$ | $\mathbf{2.46 \pm 1.05}$ | $\mathbf{23.99 \pm 4.94}$ |
| ACCEL$^{\parallel}$ | $11.30 \pm 6.37$ | $10.03 \pm 2.43$ | $59.83 \pm 4.43$ | $2.46 \pm 1.05$ | $18.29 \pm 1.69$ |
| **MEP** | | | | | |
| Random | $8.0 \pm 9.12$ | $22.46 \pm 13.34$ | $58.33 \pm 34.83$ | $2.55 \pm 2.76$ | $31.85 \pm 19.69$ |
| Stay | $4.86 \pm 7.21$ | $5.2 \pm 10.85$ | $31.55 \pm 47.13$ | $0.0 \pm 0.0$ | $1.53 \pm 3.61$ |
| Oracle | $135.07 \pm 30.27$ | $39.33 \pm 13.53$ | $138.07 \pm 10.0$ | $56.1 \pm 25.41$ | $67.86 \pm 10.89$ |
| DR | $21.18 \pm 10.27$ | $18.93 \pm 4.26$ | $56.63 \pm 10.28$ | $1.86 \pm 0.74$ | $32.19 \pm 6.23$ |
| PLR$^{\perp, \parallel}$ | $8.89 \pm 2.03$ | $20.68 \pm 1.61$ | $56.66 \pm 3.38$ | $2.38 \pm 1.13$ | $29.71 \pm 2.24$ |
| Pop. PAIRED | $\mathbf{39.02 \pm 22.74}$ | $\mathbf{20.85 \pm 5.72}$ | $\mathbf{64.51 \pm 8.74}$ | $\mathbf{2.95 \pm 2.04}$ | $\mathbf{33.29 \pm 6.11}$ |
| ACCEL$^{\parallel}$ | $12.00 \pm 6.40$ | $19.06 \pm 1.73$ | $51.94 \pm 5.28$ | $2.42 \pm 2.44$ | $28.91 \pm 1.51$ |

Table 15: Zero-shot results using SoftMoE-LSTM policies playing with an FCP and MEP population of experts trained on the respective layout exclusively. We report the *mean solved rate* and standard deviation. The best result per layout is put in **bold**.

| Method | Asymm | Counter | Cramped | Forced | Coord |
|---|---|---|---|---|---|
| **FCP** | | | | | |
| Random | $8.52 \pm 17.52\%$ | $5.00 \pm 6.70\%$ | $69.43 \pm 38.45\%$ | $0.00 \pm 0.00\%$ | $30.89 \pm 3.83\%$ |
| Stay | $6.81 \pm 18.04\%$ | $0.02 \pm 0.14\%$ | $21.75 \pm 33.71\%$ | $0.00 \pm 0.00\%$ | $0.14 \pm 0.74\%$ |
| Oracle | $69.67 \pm 16.39\%$ | $27.39 \pm 19.02\%$ | $31.30 \pm 20.97\%$ | $92.02 \pm 1.19\%$ | $96.96 \pm 2.23\%$ |
| DR | $25.99 \pm 13.37\%$ | $6.49 \pm 4.65\%$ | $67.85 \pm 8.57\%$ | $0.03 \pm 0.05\%$ | $30.26 \pm 21.34\%$ |
| PLR$^{\perp,\parallel}$ | $10.95 \pm 3.10\%$ | $3.82 \pm 2.18\%$ | $68.11 \pm 1.83\%$ | $0.07 \pm 0.05\%$ | $25.99 \pm 7.39\%$ |
| Pop. PAIRED | $\mathbf{60.18 \pm 18.29}\%$ | $\mathbf{9.08 \pm 4.51}\%$ | $\mathbf{77.71 \pm 83.7}\%$ | $0.21 \pm 0.30\%$ | $\mathbf{35.16 \pm 11.97}\%$ |
| ACCEL$^{\parallel}$ | $13.84 \pm 10.17\%$ | $7.11 \pm 3.77\%$ | $67.26 \pm 2.48\%$ | $\mathbf{0.31 \pm 0.68}\%$ | $21.33 \pm 3.81\%$ |
| **MEP** | | | | | |
| Random | $9.25 \pm 2.02\%$ | $36.04 \pm 4.38\%$ | $67.75 \pm 5.48\%$ | $0.00 \pm 0.00\%$ | $54.9 \pm 5.55\%$ |
| Stay | $4.91 \pm 1.46\%$ | $5.85 \pm 2.71\%$ | $29.56 \pm 5.92\%$ | $0.00 \pm 0.00\%$ | $1.02 \pm 0.51\%$ |
| Oracle | $91.02 \pm 1.12\%$ | $52.60 \pm 11.37\%$ | $96.86 \pm 2.27\%$ | $56.16 \pm 21.85\%$ | $75.23 \pm 0.91\%$ |
| DR | $29.51 \pm 17.44\%$ | $28.18 \pm 7.52\%$ | $65.81 \pm 8.62\%$ | $0.10 \pm 0.09\%$ | $51.61 \pm 6.33\%$ |
| PLR$^{\perp,\parallel}$ | $10.26 \pm 3.43\%$ | $\mathbf{31.81 \pm 4.53}\%$ | $65.71 \pm 3.43\%$ | $0.26 \pm 0.32\%$ | $\mathbf{49.97 \pm 3.64}\%$ |
| Pop. PAIRED | $\mathbf{50.39 \pm 29.73}\%$ | $30.66 \pm 12.23\%$ | $\mathbf{74.21 \pm 7.05}\%$ | $0.29 \pm 0.49\%$ | $45.16 \pm 17.49\%$ |
| ACCEL$^{\parallel}$ | $14.16 \pm 10.53\%$ | $27.36 \pm 4.51\%$ | $63.61 \pm 4.46\%$ | $\mathbf{0.83 \pm 1.81}\%$ | $49.71 \pm 2.32\%$ |

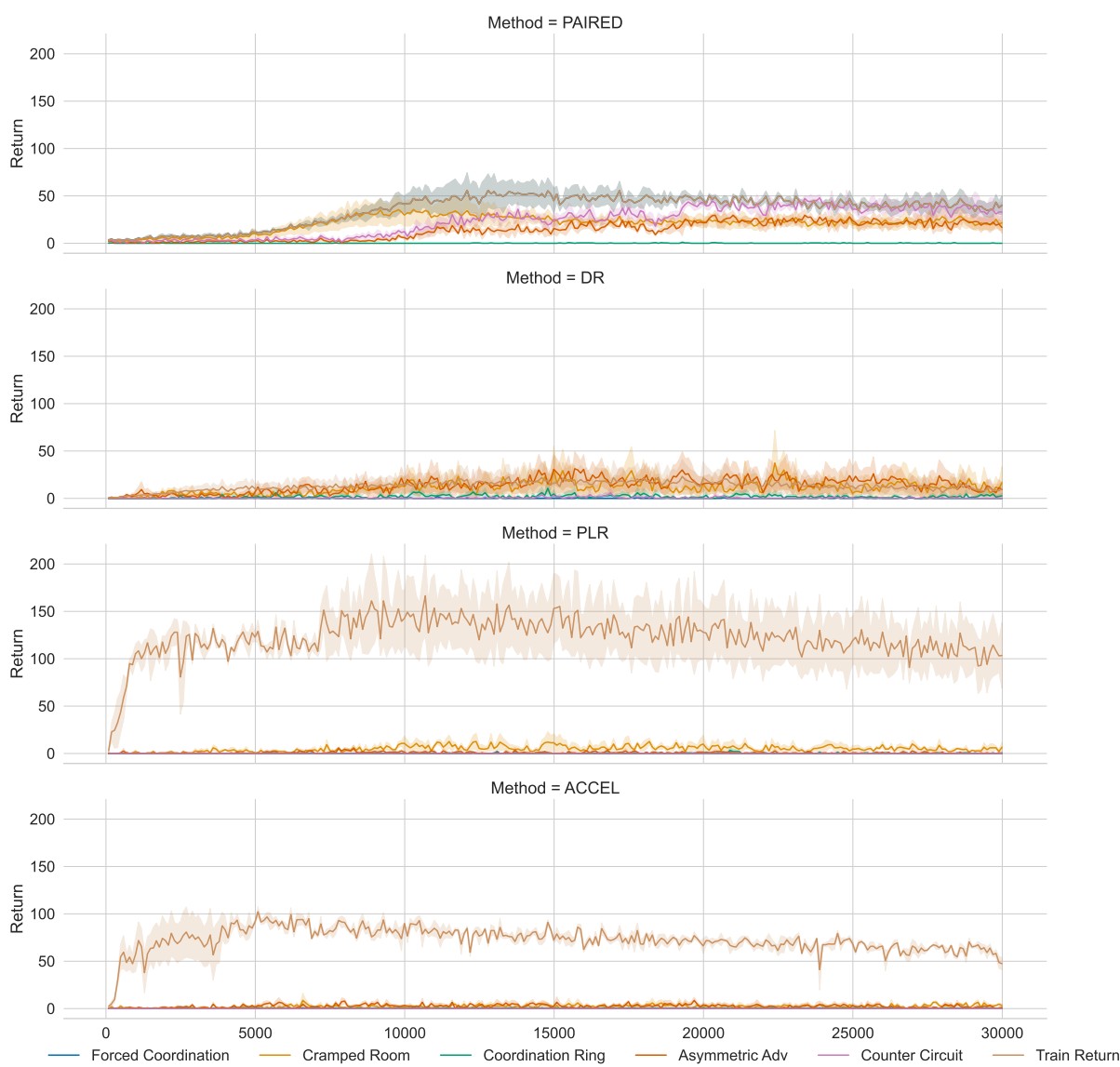

Figure 15: Returns in training and evaluation levels over the duration of training for our **SoftMoE** architecture.

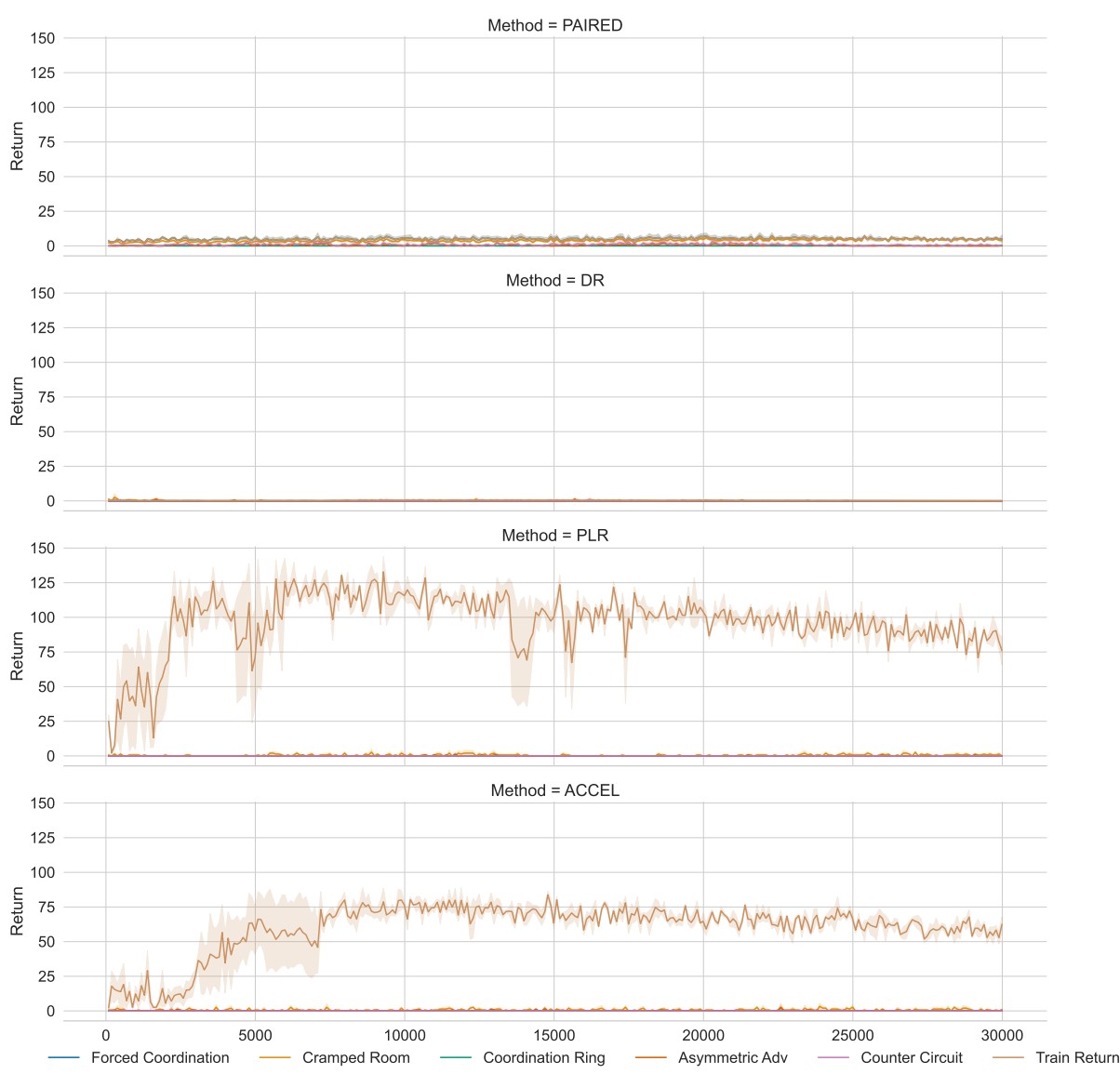

Figure 16: Returns in training and evaluation levels over the duration of training for our **S5** architecture.

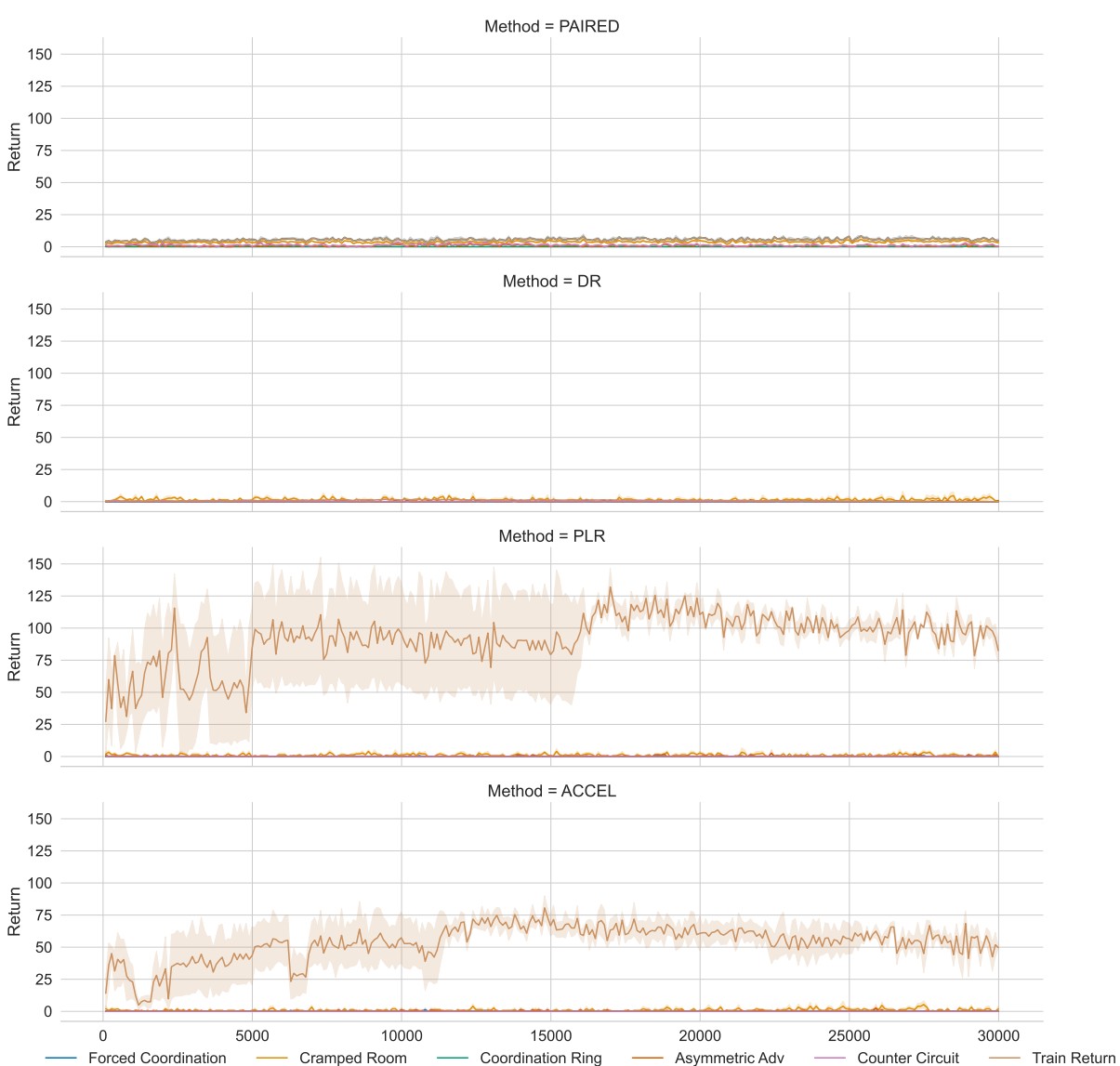

Figure 17: Returns in training and evaluation levels over the duration of training for our **CNN-LSTM** architecture.

