# OpenReview forum: "The Overcooked Generalisation Challenge: Evaluating Cooperation with Novel Partners in Unknown Environments Using Unsupervised Environment Design"
_TMLR — Accepted by TMLR_

### Review · Reviewer_AnxK · 2025-06-21

**Summary Of Contributions:**

This paper presents the Overcooked Generalization Challenge (OGC), a benchmark that addresses the critical gap in evaluating cooperative multi-agent reinforcement learning systems regarding joint environment and partner generalization. The work makes three core contributions:
- Unlike prior work that augments fixed test environments, OGC uses unsupervised environment design (UED) to procedurally generate diverse training layouts and then evaluates agents in entirely novel environments with unknown partners.
- Development of a JAX-accelerated extension of Overcooked-AI that supports parallelization and integration with state-of-the-art dual curriculum design (DCD) algorithms.
- A systematic evaluation of multiple DCD algorithms (DR, PLR, PAIRED, ACCEL) and neural architectures (CNN-LSTM, SoftMoE-LSTM, CNN-S5), demonstrating that current state-of-the-art methods largely fail at joint generalization.

**Audience:**

Yes

**Claims And Evidence:**

Yes

**Requested Changes:**

Suggested changes (non-critical):
- The paper assumes familiarity with the Overcooked-AI environment but would benefit from a brief description to provide necessary context for readers who are unfamiliar with it and its challenges.
- The introduction lacks a clear definition of Dual Curriculum Design (DCD). A concise definition should be provided when it is first introduced.
- Please verify and update citations with proper venue information where applicable, in relation to works cited as CoRR/arXiv preprints.

Minor points (e.g., typos, personal suggestions):
- Figure 6 caption: "(middle row, leftmost and rightmost)" should capitalize "Middle" if referring specifically to the "Middle" row label.
- Figure 8: "Counter Cicuit" -> "Counter Circuit"
- A.4: "12,800,00 steps" -> "12,800,000 steps"
- A.6.2: "solved rate on display it in Table 12" is poorly phrased.

**Strengths And Weaknesses:**

The authors present a well-executed and methodologically rigorous contribution that addresses a fundamental gap in cooperative multi-agent evaluation. The benchmark design is particularly strong, as it thoughtfully combines environment and partner generalization into a unified framework that better reflects real-world collaboration scenarios. The experimental methodology is comprehensive and well-justified, featuring careful selection of baseline policies and architectures alongside systematic evaluation across multiple coordination settings. The empirical analysis provides valuable insights that challenge assumptions from simpler domains, demonstrating that results from navigation-based tasks do not transfer to complex cooperative environments. The authors effectively identify specific failure modes and provide clear explanations for why current methods struggle, particularly highlighting agents' inability to learn spatially invariant coordination strategies. Overall, the paper makes a compelling case for the necessity of joint environment-partner generalization and establishes clear directions for future research through the proposed UED-ZSC framework.

---

> ### Author Response · Authors · 2025-06-26
>
> Dear AnxK,
>
> Thank you for your thoughtful and positive review.
> We appreciate your recognition of our contributions and the detailed feedback.
> We will incorporate your suggested changes in the next revision of the paper, which will incorporate feedback from all reviews once they are available.
>
> Thanks, The Authors

---

> ### Author Response · Authors · 2025-07-02
>
> Dear AnxK,
>
> Thank you again for your thoughtful and constructive review. We have now implemented all your suggestions in our latest revision (marked in blue for visibility). Specifically:
>
> > The paper assumes familiarity with the Overcooked-AI environment but would benefit from a brief description to provide necessary context for readers who are unfamiliar with it and its challenges.
>
> We added a description of the Overcooked environment in Section 4.
>
> > The introduction lacks a clear definition of Dual Curriculum Design (DCD). A concise definition should be provided when it is first introduced.
>
> We have added a brief definition to the introduction.
>
> > ​​Please verify and update citations with proper venue information where applicable, in relation to works cited as CoRR/arXiv preprints.
>
> Thanks for the attention to detail. We fixed these.
>
> > Minor points (e.g., typos, personal suggestions)
>
> Fixed, thanks for pointing them out.
>
> Thanks, The Authors

---

### Review · Reviewer_vxd9 · 2025-06-29

**Summary Of Contributions:**

The paper introduces the “overcooked generalization challenge (OGC)”, a new benchmark for generalizable, cooperative multi-agent RL. Building on overcooked AI the challenge focuses on generalization capabilities to both new environments (i.e., kitchen layouts) and partners. For training, randomly generated environments can be used but the framework provides interfaces for unsupervised environment design and curriculum design.
The paper also provides an empirical evaluation of existing SOTA approaches on the new benchmark.

**Audience:**

Yes

**Broader Impact Concerns:**

No concerns

**Claims And Evidence:**

Yes

**Requested Changes:**

Major: Run experiments for more seeds and provide statistically robust uncertainty measures

Minor: Provide some additional analysis of why SOTA fails.

**Strengths And Weaknesses:**

The benchmark addresses the very important problem of generalization in human-AI collaboration, which is crucial for future real-world approaches. It's technically well executed by building on existing tasks (overcooked) and frameworks (minmax). Also, the framework is built with scalability in mind (highly parallelized JAX implementation) and provides interfaces for different forms of DCD (proposing entirely new levels and modifying a given level). Overall, this results in a highly accessible, useful, and valuable benchmark.

Additionally, the empirical evaluation shows that the proposed benchmark is indeed beyond the capabilities of the current SOTA and thus relevant.

One weakness of the evaluation is the number of seeds and how the results are presented. The paper uses 3 seeds and reports the mean and standard deviations. While this will most likely not change the overall finding that SOTA approaches struggle with the challenge, an appropriate number of seeds (>=10) and statistically more robust quantities (e.g., confidence intervals) should be provided. This would also allow future publications to directly “quote” the results from the paper, making the benchmark more accessible and the findings generated with it more significant.

Additionally, while the work's main focus is establishing the benchmark, not a solution, I would appreciate further insights into why SOTA methods fail. For example, one point I was wondering about is the ratio of solvable vs unsolvable environments generated by the curriculum (in particular, the random one. Intuitively, I would suspect that just placing everything randomly does not give many valid environments.) This could already indicate if the issue is more on the curriculum generation side or actually on the policy learning side?

---

> ### Author Response · Authors · 2025-07-02
>
> Dear vxd9,
>
> Thank you very much for your constructive and encouraging review. We’re especially grateful for your recognition of the importance, accessibility, and scalability of the Overcooked Generalisation Challenge (OGC). Please find our answers below.
>
> > Number of seeds and confidence intervals
>
> We appreciate this suggestion and agree that reporting results over more seeds would improve the statistical robustness and reproducibility of our findings. As you also noted, however, we also don’t expect significantly different results from an extended evaluation. That being said, while our JAX-based implementation is parallelisable and fast, training across many seeds remains computationally expensive, mostly due to the overhead introduced by online environment generation and curriculum learning.
> As such, running ≥10 seeds per configuration (across four UED methods × three architectures and for baselines) is currently beyond our available compute.
>
> We are currently running evaluations in which we increased the number of seeds per method to 5, which aligns with prior Overcooked-AI work [1–4]. Following your suggestion, we will also report 95% confidence intervals alongside standard deviations to facilitate the statistical interpretation of our results. We will update the submission with the results when they are available. We hope this sufficiently addresses your concern regarding statistical robustness.
>
> [1] Carroll, Micah, et al. "On the utility of learning about humans for human-ai coordination." Advances in neural information processing systems 32 (2019).
>
> [2] Zhao, Rui, et al. "Maximum entropy population-based training for zero-shot human-ai coordination." Proceedings of the AAAI Conference on Artificial Intelligence. Vol. 37. No. 5. 2023.
>
> [3] Yan, Xue, et al. "An efficient end-to-end training approach for zero-shot human-AI coordination." Advances in Neural Information Processing Systems 36 (2023): 2636-2658.
>
> [4] Strouse, D. J., et al. "Collaborating with humans without human data." Advances in Neural Information Processing Systems 34 (2021): 14502-14515.
>
>
> > Ratio of solvable vs unsolvable environments generated by the curriculum
>
> Thank you for this suggestion. While there is no established method for formally proving layout solvability, we agree this is an important direction for understanding agent failure modes. In the revised version of the paper, we report the empirical solvability rate based on manual inspection of 200 layouts sampled from the random environment generator. We find that 74,5% of levels are solvable when sampled from the random generator (inaccessible items, etc.). Among the solvable layouts, roughly half are of comparable difficulty to the original Overcooked-AI layouts. This estimate is based on qualitative visual inspection of sampled environments, where we assessed factors such as object placement, agent accessibility, and the presence of coordination bottlenecks. The remaining layouts, while technically solvable, appeared to require significantly more complex or inefficient action sequences (e.g., very long detours, multiple or long tight corridors, asymmetric access to key stations, dependence on difficult item handover paths). This highlights that generating feasibly solvable levels is itself a key challenge posed by the OGC.
>
> Thanks, The Authors

---

> > ### Author Response · Authors · 2025-07-12
> >
> > Dear Reviewer,
> >
> > Thanks again for the positive feedback and your review. As encouraged we have increased the number of seeds per method to five for the reasons outlined above and now also report the 95% confidence interval. As also predicted by you, no conclusions or insights change. You can find these in our uploaded revision. We marked the changes in blue in the text. We hope that this satisfies your request. Please note that we will update the numerous tables in the appendix in the camera ready version.
> >
> > Thanks, the Authors

---

> > > ### Comment · Reviewer_vxd9 · 2025-07-15
> > >
> > > Dear Authors,
> > >
> > > Thank you for addressing my concerns. I have no further comments.

---

### Review · Reviewer_EvB3 · 2025-06-30

**Summary Of Contributions:**

This paper introduces the Overcooked Generalisation Challenge (OGC), the first benchmark specifically designed to evaluate reinforcement learning agents' ability to perform zero-shot cooperation with novel partners in previously unseen environments. This benchmark requires simultaneous generalization across both environments and partners. The authors also contribute OvercookedUED, a GPU-accelerated, open-source environment built on JAX that integrates with state-of-the-art dual curriculum design (DCD) methods. They provide comprehensive baseline evaluations using current DCD algorithms (PAIRED, PLR, ACCEL, DR) combined with modern neural architectures (CNN-LSTM, SoftMoE-LSTM, CNN-S5). The empirical results reveal that even state-of-the-art methods struggle with both novel environment and partner generalization, highlighting the difficulty of the proposed benchmark and the limitations of current approaches.

**Audience:**

Yes

**Claims And Evidence:**

Yes

**Requested Changes:**

The authors may consider formalizing additional metrics to capture the cooperation efficiency to better examine the generalization with novel partners.

**Strengths And Weaknesses:**

__Strengths:__

* The motivation for this work, studying out-of-distribution generalization in cooperative settings, is compelling and well-articulated.
* The GPU-accelerated implementation of the testbed environment using JAX, achieving up to 2M steps per second, makes large-scale experiments feasible. The integration with the minimax framework ensures compatibility with existing DCD methods.
* The authors provide qualitative failure analysis, providing insights into the limitations of existing approaches.

__Weaknesses:__

* The proposal of the benchmark lacks a theoretical analysis of why environment and partner generalization should be coupled, or what fundamental properties make this problem difficult.
* The metrics used in evaluating cooperative tasks, such as average return and solvable rates, are not direct and accurate indicators of agent collaboration efficiency and activeness.
* The five hand-designed layouts for evaluation may not adequately represent the full spectrum of cooperative challenges.

__Questions:__

* This benchmark looks specific for DCD algorithm benchmarking. Could this benchmark be used for other generalization methods in RL, such as meta-learning, domain adaptation, and invariant representation learning?

---

> ### Author Response · Authors · 2025-07-02
>
> Dear EvB3,
>
> Thank you for your thoughtful and detailed review. We are grateful for your recognition of the motivation, technical contributions, and baseline analysis presented in the Overcooked Generalisation Challenge (OGC). We address your comments and suggestions below.
>
> > Theoretical analysis of why environment and partner generalization should be coupled
>
> We have expanded our discussion in the introduction to highlight structural dependencies between the environment and partner behaviour. For example how spatial bottlenecks constraints viable joint policies.
>
> > Metrics used in evaluating cooperative tasks, such as average return and solvable rates, are not direct and accurate indicators of agent collaboration efficiency and activeness.
>
> We believe that the metrics used in our evaluation (particularly average episode return) are meaningful indicators of both task efficiency and partner adaptability. Since Overcooked-AI rewards agents for delivering soups within a fixed time horizon, the total return directly reflects the agents’ ability to coordinate efficiently under time constraints.
> Moreover, zero-shot return with novel partners serves as a practical measure of adaptability to unfamiliar behaviour, which is a central challenge in ad-hoc teamwork. These metrics are standard in the cooperative RL literature and are typically used throughout prior work in Overcooked-AI [1–4].
>
> We have clarified this connection in Section 4.3. of the revised paper, and we hope this addresses the reviewer’s concern.
>
> [1] Carroll, Micah, et al. "On the utility of learning about humans for human-ai coordination." Advances in neural information processing systems 32 (2019).
>
> [2] Zhao, Rui, et al. "Maximum entropy population-based training for zero-shot human-ai coordination." Proceedings of the AAAI Conference on Artificial Intelligence. Vol. 37. No. 5. 2023.
>
> [3] Yan, Xue, et al. "An efficient end-to-end training approach for zero-shot human-AI coordination." Advances in Neural Information Processing Systems 36 (2023): 2636-2658.
>
> [4] Strouse, D. J., et al. "Collaborating with humans without human data." Advances in Neural Information Processing Systems 34 (2021): 14502-14515.
>
> > The five hand-designed layouts for evaluation may not adequately represent the full spectrum of cooperative challenges.
>
> While we include the five original Overcooked layouts due to their wide use and recognizability in the community, our evaluation goes significantly beyond them. As described in Figure 4 and Section 4.3, we also evaluate environment generalisation using 32 additional, human-authored layouts that cover a broad range of coordination structures and spatial configurations (see Section 5.2 and Table 2). These are specifically designed to test generalisation beyond the canonical layouts.
>
> > Could this benchmark be used for other generalization methods in RL, such as meta-learning, domain adaptation, and invariant representation learning?
>
> Yes! We believe that our benchmark can be used widely and is compatible with other generalisation frameworks. Although our baseline evaluation focuses on DCD methods, future works could establish other use cases. We personally are really interested in few-shot adaptation to partners and environments similar to the Ada line of work (e.g. [5]). We have clarified this in section 7.
>
> [5] Bauer, Jakob, et al. "Human-timescale adaptation in an open-ended task space." International Conference on Machine Learning. PMLR, 2023.
>
> > Requested Change: ​​The authors may consider formalizing additional metrics to capture the cooperation efficiency to better examine the generalization with novel partners.
>
> Please see our response above regarding our use of reward-based metrics as indicators of coordination efficiency and adaptability. Does the reviewer have additional other metrics in mind that they would be interested in?
>
> Thanks, the Authors

---

> > ### Comment · Reviewer_EvB3 · 2025-07-13
> >
> > I thank the authors for their detailed response. I have no other concerns.

---

### Decision · Action_Editor_QpS2 · 2025-08-07

**Recommendation:** Accept as is

**Audience:**

Yes

**Audience Explanation:**

This benchmark would be of great use to the multi-agent RL community.

**Claims And Evidence:**

Yes

**Claims Explanation:**

The authors introduce the Overcooked Generalization challenge, which includes an open-source GPU-accelerated implementation. They provide both quantitative and qualitative evidence for its usefuleness and effectiveness, which lends credibility to the proposal.